# Safety Aware Changepoint Detection for Piecewise i.i.d. Bandits

**Subhojyoti Mukherjee**[1]

[1]Electrical & Computer Engineering Department, UW-Madison, Madison, Wisconsin, USA

## Abstract

In this paper, we consider the setting of piecewise i.i.d. bandits under a safety constraint. In this piecewise i.i.d. setting, there exists a finite number of changepoints where the mean of some or all arms change simultaneously. We introduce the safety constraint studied in Wu et al. [2016] to this setting such that at any round the cumulative reward is above a constant factor of the default action reward. We propose two actively adaptive algorithms for this setting that satisfy the safety constraint, detect changepoints, and restart without the knowledge of the number of changepoints or their locations. We provide regret bounds for our algorithms and show that the bounds are comparable to their counterparts from the safe bandit and piecewise i.i.d. bandit literature. We also provide the first matching lower bounds for this setting. Empirically, we show that our safety-aware algorithms perform similarly to the state-of-the-art actively adaptive algorithms that do not satisfy the safety constraint.

## 1 INTRODUCTION

Consider a startup XYZ that wants to maximize revenue collection from ad placements when users land on their webpage. Revenue is generated when users click on the ads. The user preferences change over time but not quickly enough so that XYZ can focus on maximizing the revenue collection for some time before changing its strategy again. To do this XYZ must detect the user's new preferences and modify its suggestions. Due to budget constraints XYZ must make sure that the aggregate revenue collection must not fall below a certain threshold. The difficulty is that XYZ does not know placing which ads will surely result in revenue above the threshold. This constrains XYZ from randomly placing different ads on their landing webpage.

On consultation with the industry experts XYZ comes up with a default action that is known historically to be a highly favored by users. Hence, XYZ comes up with a new safety constraint such that when their algorithm is unsure of which ad to place for some user it can fall back on this default action. The learning algorithm now has to balance between exploration under safety constraints and exploitation in this slowly changing environment.

The dilemma faced by XYZ can be modeled as a sequential decision making problem in the piecewise i.i.d. bandit setting under safety constraints. In the piecewise i.i.d. bandit setting the learner is provided with a set of arms $i \in \{0, 1, 2, \ldots, K\}$ where we index the default arm (baseline) as $i = 0$, and there exists a finite number of changepoints where the mean $\mu_i$ of one or more arms may change simultaneously. At every round $s \in \{1, 2, \ldots, T\}$ the learner selects an action $I_s \in \{0, 1, 2, \ldots, K\}$ and observes the feedback $X_{I_s}(s)$ where $\mathbb{E}[X_{I_s}(s)] =: \mu_{I_s}(s)$. Define $i^*$ as the optimal arm such that $\mu_{i^*}(s) > \mu_i(s)$ for all $i$. The goal of the learner is to maximize reward by quickly finding the optimal arm $i^*$ under the following safety constraint

$$\sum_{s=1}^{t} X_{I_s}(s) \geq (1 - \alpha) \sum_{s=1}^{t} X_0(s) \qquad (1)$$

for all $t \in \{1, \ldots, T\}$, where $\alpha \in (0, 1]$ is the risk parameter, and $T$ is the horizon. The constraint in eq. (1) represents how much the learner is allowed to risk in conducting the exploration. For example, if $\alpha \to 0$ the learner is expected to sample arms that are at least better than the baseline arm 0. The baseline arm represents expert's belief over the current user preferences and may change over time. Similar to the setting of Wu et al. [2016] we assume that the mean of the baseline arm is not known to the learner. Note that Wu et al. [2016] is not suited for the piecewise i.i.d setting. By change of belief of the expert we mean that only the value of the baseline arm changes and not the index.

The challenge in our setting is three-fold: **1)** Ensure that the safety constraint (1) on cumulative reward is satisfied.

*Accepted for the 38th Conference on Uncertainty in Artificial Intelligence* (UAI 2022).

Consider the scenario where the risk parameter $\alpha = 1$. In this case satisfying the eq. (1) is easy as choosing any arm satisfies the constraint. However as $\alpha \to 0$ maintaining the safety constraint becomes difficult as exploration becomes limited or it will violate the safety constraints. **2)** Adapt to the piecewise i.i.d. nature of the environment. Observe that as the means of arms change abruptly at changepoints the algorithm must adapt or the safety constraint will be violated. Further, to detect changepoints the algorithm must conduct additional exploration without violating the eq. (1). Note that Wu et al. [2016] do not consider any such piecewise i.i.d. setting. **3)** Finally, minimize the cumulative regret by quickly finding the optimal arm for each of the time segment between two changepoints. Our contributions are as follows:

1) We formulate the novel piecewise i.i.d. bandit setting under safety constraints. We show that the current state-of-the-art conservative algorithms [Wu et al., 2016] as well as the changepoint detection algorithms are not equipped to handle the safety constraint in eq. (1) in this setting.

2) We propose two actively adaptive algorithms that detect changepoints and restart by erasing past history of interactions. Simultaneously these algorithms ensure that the safety constraint is satisfied. The current changepoint detection algorithms [Besson and Kaufmann, 2019, Mukherjee and Maillard, 2019, Besson et al., 2020] do not take into account the safety constraint and hence are not suited for our setting.

3) We provide theoretical guarantees for both of our algorithms and uncover new problem dependent terms that depends on the optimality gaps, changepoint gaps, the gap of the baseline arm, and the risk parameter $\alpha$. We also provide the first matching lower bounds for this setting. Empirically we show that our proposed methods perform comparably against safety oblivious changepoint detection algorithms.

## 2  RELATED WORKS

Our work lies at the intersection of two interesting areas: 1) Changepoint detection in piecewise i.i.d bandits, and 2) Safe Sequential Decision Making. In piecewise i.i.d bandits it is assumed that the change of mean (drift) of an arm are well separated and significant enough to be detected. The previous works in this setting are broadly classified into two groups, viz. passively adaptive and actively adaptive algorithms. Passively adaptive algorithms such as Discounted UCB (D-UCB) [Kocsis and Szepesvári, 2006], Sliding Window UCB (SW-UCB) [Garivier and Moulines, 2011], and Discounted Thompson Sampling (D-TS) [Raj and Kalyani, 2017] do not try to detect the changepoints and only focus on minimizing the regret over a short window of the time horizon. On the contrary, actively adaptive algorithms such as EXP3.R [Allesiardo et al., 2017], CD-UCB [Liu et al., 2017], CUSUM [Liu et al., 2017], M-UCB [Cao et al., 2018], GLR-UCB [Besson and Kaufmann, 2019, Besson et al., 2020], Ad-Switch [Auer et al., 2019], and UCB-CPD

[Mukherjee and Maillard, 2019] try to detect the changepoints and restart by erasing all the past history of interactions. Actively adaptive algorithm like GLR-UCB, M-UCB, UCB-CPD has several advantages over passively adaptive algorithms. In environments where the changepoint gaps are large and well-separated the passively adaptive algorithms perform poorly (see [Besson and Kaufmann, 2019]). The EXP3.R (an adaptive version of EXP3.S [Auer et al., 2002b]) is more pessimistic than other actively adaptive algorithms like UCB-CPD, GLR-UCB as it uses the conservative exponential weighting algorithm EXP3 for changepoint detection, and hence, performs poorly in practice. GLR-UCB uses the Bernoulli generalized likelihood ratio test involving Kullback Leibler (KL) based divergence function as changepoint detector. The KL divergence function of GLR-UCB better exploits the geometry of (sub-)Bernoulli distributions and so it outperforms M-UCB, Ad-Switch. Note that none of the above algorithms are safety aware.

The safe sequential decision making setup has recently garnered a lot of attention in machine learning [Amodei et al., 2016, Turchetta et al., 2019]. Closer to our setting are the works that study regret minimization in bandits under safety constraints such as Wu et al. [2016], Kazerouni et al. [2017], Amani et al. [2019], Garcelon et al. [2020]. These works encode their safety requirements in the form of constraints on the cumulative rewards observed by the learner. This setup also called conservative bandits as the exploration is constrained by the constraints on the cumulative reward. Note that while Wu et al. [2016] studies the unstructured stochastic and adversarial bandit setting, the Kazerouni et al. [2017], Amani et al. [2019], Garcelon et al. [2020] study the linear bandit (structured) setting. Another line of related work [Moradipari et al., 2020, Pacchiano et al., 2020, Khezeli and Bitar, 2020] focuses on the idea of stagewise safety constraint where at every stage (round) the reward should be higher than a predetermined safety threshold with high probability. Note that our setting of safety constraints on cumulative rewards cannot be directly applied to the stagewise setting. None of the above works deal with the setting of piecewise i.i.d bandits with slow drift (change of means). Note that while Wu et al. [2016] studies an adversarial setting, they do not take any assumption on the reward distributions and hence their exploration scheme is highly conservative for piecewise i.i.d. bandits. Similarly, conservative bandits studying the contextual (structured) bandit setting [Kazerouni et al., 2017] do not use the known information about slow drift of means. Finally note that our setting is different than the thresholding bandit problem [Locatelli et al., 2016, Mukherjee et al., 2017] where the goal is to find all the arms above a fixed threshold $B$ under the fixed budget setting.

**Notations:** Denote $[n] := \{1, 2, \ldots, n\}$. Define the set of arms as $[K]$ indexed from $i = 1, 2, \ldots, K$. The baseline arm is denoted by the index $0$. Note that the learner knows

the index of the baseline arm but it does know the mean of the baseline arm. This is similar to the setting in Wu et al. [2016]. We define the set $[K]^+ := [K] \cup \{0\}$ to indicate that baseline arm 0 is included as well. We define the mean of the arm $i$ at round $s$ as $\mu_i(s)$, and the empirical mean of the arm till round $t$ as $\widehat{\mu}_i(t)$. We denote the optimal arm as $i^*$ and the mean of the optimal arm at round $s$ as $\mu_{i^*}(s)$. The reward of the arm $i$ sampled at round $s$ is denoted by $X_i(s)$. We assume that the rewards are coming from a bounded distribution supported on $[0, 1]$. We further denote the distribution of the $i$-th arm with mean $\mu_i(t)$ as $\nu(\mu_i(t))$. We denote the horizon (total rounds) as $T$. The safety threshold is denoted by $B$ and the risk parameter is denoted by $\alpha \in [0, 1]$. For brevity we denote the sequence of rounds between $s$ to $t$ as $s : t$. For clarity of presentation we overload the notation $\mu_i(\cdot)$ to denote either the mean at round $s$ as $\mu_i(s)$ or the mean over the rounds $1 : t$ as $\mu_i(1 : t)$. Similarly, $N(1 : t)$ denotes that number of pulls of $i$ from $1 : t$. Define the empirical mean $\widehat{\mu}_i(1 : t) := \frac{\sum_{s=1}^t X_{I_s} \mathbb{I}\{I_s = i\}}{\sum_{s=1}^t \mathbb{I}\{I_s = i\}} = \frac{\sum_{s=1}^t X_{I_s} \mathbb{I}\{I_s = i\}}{N_i(1:t)}$ where $I_s$ denotes the arm pulled at round $s$.

# 3 GLOBAL CHANGEPOINT DETECTION

We now define the setup for the Global Changepoint Setting (GCS). Let the total number of changepoints till round $T$ be denoted by $G_T$ such that

$$G_T := \# \left\{ 1 \leq s \leq T \mid \exists i \in [K]^+ : \mu_i(s-1) \neq \mu_i(s) \right\}. \tag{2}$$

We define the global changepoints $t_{c_0} < t_{c_1} < t_{c_2} < \ldots < t_{c_{G_T}}$ such that the $g$-th global changepoint is defined as:

$$t_{c_g} := \inf\{s > t_{c_{g-1}} : \forall i \in [K]^+, \mu_i(s-1) \neq \mu_i(s)\}.$$

Hence at a global changepoint $t_{c_g}$ the mean of all the arms $i \in [K]^+$ change simultaneously. Let $t_{c_0} = 1$ by convention. Note, that the baseline mean changes at changepoints to signify the new belief of the experts based on updated user preferences. We define the changepoint segment between $t_{c_g}$ to $t_{c_{g+1}} - 1$ as $\rho_g$. Note that the optimal arm for each changepoint segment $\rho_g$ may or may not be same. We further define a few notations. Let the confidence width of arm $i$ for rounds $1 : t$ be defined as

$$\beta_i(1 : t, \delta) = \sqrt{\frac{2 \log(4 \log_2(t+1)/\delta)}{N_i(1:t)}} \tag{3}$$

with the standard condition that if $N_i(1 : t) = 0$ then $\beta_i(1 : t, \delta) = \infty$. In our case it suffices to take the leading constant of $\beta_i(1 : t, \delta)$ as 2, though tighter bounds are known and can be used in practice, e.g. Balsubramani [2014], Tanczos et al. [2017], Howard et al. [2021]. These type of anytime bounds constructed with $\beta_i(1 : t, \delta)$ are known to be tight in the sense

that $\mathbb{P}\left(\bigcup_{t=1}^\infty \left\{|\widehat{\mu}_i(1:t) - \mu_i(1:t)| \geq \beta_i(1:t,\delta)\right\}\right) \leq \delta$ and that there exists an absolute constant $C \in (0,1)$ such that $\mathbb{P}\left(\left\{|\widehat{\mu}_i(1:t) - \mu_i(1:t)| \geq C\beta_i(1:t,\delta) \text{ for infinitely many } t \in \mathbb{N}\right\}\right) = 1$ by the Law of the Iterated Logarithm [Hartman and Wintner, 1941]. Next we define the upper confidence bound for $i$ as

$$U_i(1:t) := \widehat{\mu}_i(1:t) + \beta_i(1:t,\delta) \tag{4}$$

and the lower confidence bound from $1 : t$ as

$$L_i(1:t) := \widehat{\mu}_i(1:t) - \beta_i(1:t,\delta). \tag{5}$$

We define the UCB arm $u_t$ at round $t$ as

$$u_t := \arg\max_{i \in [K]} U_i(1:t) \tag{6}$$

which is the arm with the highest uncertainty and needs to be explored more to get a better estimate of its true mean [Agrawal, 1995, Auer et al., 2002a]. Finally, we define the empirical safety budget as

$$\widehat{Z}(1:t) := \sum_{s=1}^{t-1} L_{I_s}(1:s) + L_{u_t}(1:t) - (1-\alpha)\sum_{s=1}^t U_0(1:t) \tag{7}$$

which quantifies by how much the safety constraint is being violated. Also recall that the baseline arm is indexed as 0, and the learner does not know the mean of the baseline arm. This is similar to the second setting in Wu et al. [2016].

## 3.1 SAFE GLOBAL RESTART ALGORITHM

In this section we introduce the Safe Global Restart (SGR) algorithm which is a safety aware global changepoint detection algorithm. SGR is an actively adaptive algorithm and so it restarts by erasing the history of interactions once it detects a changepoint. We define the parameter $r_i$ for the $i$-th arm as the last restart round when a changepoint was detected and the arm $i$ history was erased. We define the last restart vector

$$\boldsymbol{r} := \{r_1, r_2, \ldots, r_K\} \cup \{r_0\}.$$

The safety budget for the GCS is $\widehat{Z}(1 : t)$

$$:= \sum_{s=1}^{t-1} L_{I_s}(1:s) + U_{u_t}(1:t) - (1-\alpha)\sum_{s=1}^t U_0(1:t)$$

$$\overset{(a)}{=} \sum_{s=1}^{t-1} L_{I_s}(r_{I_s}:s) + U_{u_t}(r_{u_t}:t) - (1-\alpha)\sum_{s=1}^t U_0(r_0:t) \tag{8}$$

where, $(a)$ follows because when a changepoint is detected the history is erased for that arm $i \in [K^+]$.

We now state the main aspects of SGR . SGR is initialized by sampling each arm once. Then at every round SGR decides to pull the UCB arm $u_t$ if $\widehat{Z}(1 : t) \geq 0$ or the baseline arm 0

if $\widehat{Z}(1:t) < 0$. Then SGR samples the next arm, observes the reward $X_{I_t}(t)$ and updates the problem parameters. Finally SGR calls the CPD changepoint detector sub-routine to detect any changepoint. If a changepoint is detected at round $t$ by CPD then it erases the history of interactions for all arms (including baseline arm) and sets the restarting time for all arms $i \in [K]^+$ as $r_i = t$. We state the pseudo-code of the policy SGR in Algorithm 1 and the key idea behind CPD in the following Section 3.2.

---

**Algorithm 1** Safe Global Restart (SGR)

---

1: **Input:** Risk parameter $\alpha \in [0, 1)$
2: Set $r_i = 1, \forall i \in [K]^+$. Pull each arm once.
3: **for** $t = K^+ + 1, K^+ + 2, \ldots$ **do**
4:     **if** $\widehat{Z}(t) \geq 0$ **then**
5:         Set $I_t = u_t$ from eq. (6)     ▷*Pull UCB arm*
6:     **else if** $\widehat{Z}(t) < 0$ **then**
7:         Set $I_t = 0$     ▷*Baseline arm*
8:     Pull $I_t$ and observe $X_{I_t}(t)$.
9:     Update $\widehat{\mu}_{I_t}(r_{I_t} : t), N_{I_t}(r_{I_t} : t)$, and $\widehat{Z}(r_{I_t} : t)$ in eq. (7).
10:     Call CPD $(\boldsymbol{r}, t, \text{global})$     ▷*Call CPD*

---

## 3.2 CHANGEPOINT DETECTION

The sequential changepoint detection has a long history in the statistical community [Basseville et al., 1993, Wu, 2007]. We explain the sequential changepoint detection through the following example: Consider a single arm $i$. Let at some round $t$ we have a collection of i.i.d. samples $X_i(1), X_i(2), \ldots, X_i(t)$ from a bounded distribution that is supported on $[0, 1]$. The goal of changepoint detection is to find out whether all the $t$ samples have come from the same distribution with mean $\mu_i(1 : t)$ or there exist a changepoint $\tau_{c_g} \in \mathbb{N}$ such that $X_i(1), X_i(2), \ldots, X_i(\tau_{c_g} - 1)$ have mean $\mu_i(1 : \tau_{c_g} - 1)$ while $X_i(\tau_{c_g}), X_i(\tau_{c_g} + 1), \ldots, X_i(t)$ have a different mean $\mu_i(\tau_{c_g} : t) \neq \mu_i(1 : \tau_{c_g} - 1)$. For notational convenience let us denote $\mu_i(1 : \tau_{c_g})$ as $\mu'$ and $\mu_i(\tau_{c_g} + 1 : t)$ as $\mu''$ respectively. Hence, a sequential changepoint detector is defined as a stopping time $\tau^{chg} < \infty$ that rejects the null hypothesis $\mathcal{H}_0 : (\exists \mu' : \forall i \in \mathbb{N}, \mathbb{E}[X_i] = \mu')$ in favor of the alternate hypothesis $\mathcal{H}_1 : (\exists \mu'' \neq \mu', \tau_{c_g} \in \mathbb{N} : X_i(1), X_i(2), \ldots, X_i(\tau_{c_g}) \sim \nu(\mu')$, and $X_i(\tau_{c_g} + 1), X_i(\tau_{c_g} + 2), \ldots, X_i(t) \sim \nu(\mu''))$. Previous works have studied the Generalized Likelihood Ratio Test (GLRT) to detect the changepoints using this hypothesis testing idea. Many of the previous works [Wilks, 1938, Siegmund and Venkatraman, 1995, Maillard, 2019, Besson and Kaufmann, 2019, Besson et al., 2020] that have studied this setting used Generalized Likelihood Ratio Test (GLRT) to detect changepoints. The GLRT test works as follows: We first calculate the GLRT statistic defined by

$$\log \frac{\sup_{\mu', \mu'', \tau_{c_g} < t} L\left(X_i(1), \ldots, X_i(t); \mu', \mu'', \tau_{c_g}\right)}{\sup_{\mu'} L\left(X_i(1), \ldots, X_i(t); \mu'\right)}$$

where we denote the term $L(X_i(1), \ldots, X_i(t); \mu')$ and $L(X_i(1), \ldots, X_i(t); \mu', \mu'', \tau)$ as the likelihood of the first $t$ observations under hypothesis $\mathcal{H}_0$ and $\mathcal{H}_1$ respectively. Now if the GLRT statistic crosses a threshold $\widetilde{\beta}(t, \delta)$ then it indicates that there exists a changepoint and the null hypothesis $\mathcal{H}_0$ is rejected. A similar type of test, called the CUSUM test [Page, 1954, Liu et al., 2017] has also been studied where the distributions $\nu(\mu_1)$ and $\nu(\mu_2)$ are completely known. Note that GLRT works in the case when both distributions are unknown but they come from the same canonical exponential family. A detailed discussion on this can be found in Maillard [2019].

**Confidence-based scan statistic:** An alternative to the GLRT based scan statistics is the confidence-based scan statistic that have been studied in Mukherjee and Maillard [2019]. In the confidence based scan statistic the total number of samples of arm $i$ from $1 : t$ is divided into slices, and for each slice $s$ a confidence interval is built of the form

$$\widehat{\mu}_i(1:s) \pm \beta_i(1:s, \delta) \text{ and } \widehat{\mu}_i(s+1:t) \pm \beta_i(s+1:t, \delta) \quad (9)$$

where $\beta_i(1:s, \delta)$ is from eq. (3). Now if there exists some $s$ at which the confidence intervals do *not* overlap such that

$$\tau_{c_g} := \inf\left\{ t \in \mathbb{N} : \exists i \in [K]^+, \exists s \in [1, t], |\widehat{\mu}_i(1:s) - \widehat{\mu}_i(s+1:t)| \right.$$
$$\left. > \beta_i(1:s, \delta) + \beta_i(s+1:t, \delta) \right\} \quad (10)$$

then report a changepoint at $s$. Besson et al. [2020] show that GLRT outperforms the confidence-based scan statistic as it better exploits the geometry of the Bernoulli distributions. In our work we use the confidence-based scan statistic as our goal is not to compete in the vanilla changepoint detection setting but to derive novel bounds for the safety aware piecewise i.i.d. setup proposed in this work.

Finally, we propose the CPD changepoint detector sub-routine which is similar to the UCB-CPD algorithm in Mukherjee and Maillard [2019]. The CPD takes input the restart vector $\boldsymbol{r}$, current time $t$, and the type $\in \{\text{global}, \text{local}\}$ indicating whether it is a global or a local changepoint setting. In this section we only discuss the global setting while the local setting is discussed in Section 5. The CPD divides the total rounds $r_i : t$ into $(t - r_i)$ slices for each arm $i \in [K]^+$ and then proceeds to conduct the confidence-based scan statistics as discussed in (10). If there is a disjoint slice $s$ then CPD reports a changepoint at $s$, then erases the history of interactions (including the baseline arm) and resets the restarting round counter $r_i, \forall i \in [K]^+$ to the current time $t$. We also reset the safe budget $\widehat{Z}(1 : t)$ to 0. Ideally in practice we can still continue the accrued safe budget to the next changepoint section from $\tau_{c_{g+1}} + 1$ without setting it to 0, but this makes our theoretical analysis more tedious.

**Algorithm 2** CPD $(\boldsymbol{r}, t, \text{type})$

---
1: **for** $i = 1, 2, \ldots, K^+$ **do**
2:     **for** $t' = r_i, r_i + 1, \ldots, t$ **do**
3:       **if** $L_i(r_i : t') < U_i(t' + 1 : t)$ or $U_i(r_i : t') > L_i(t' + 1 : t)$ **then**
4:         **if** type = global **then**
5:           $\{\widehat{\mu}_j(r_i : s), N_j(r_i : s)\}_{s=r_j}^{t} = \{0, 0\}_{s=r_j}^{t}$, $\forall j \in [K^+]$. Set $r_j = t, \forall j \in [K^+]$. Set $\widehat{Z}(1 : t) = 0$.
6:         **else**
7:           $\{\widehat{\mu}_i(r_i : s), N_i(r_i : s)\}_{s=r_i}^{t} = \{0, 0\}_{s=r_i}^{t}$, $r_i = t$. Set $\widehat{Z}(1 : t) = 0$.

---

### 3.3 REGRET ANALYSIS FOR SGR

We denote the time interval segment $\rho_g := [t_{c_g}, t_{c_{g+1}} - 1]$ so that the segment $\rho_g$ starts at round $t_{c_g}$ and ends at $t_{c_{g+1}-1}$. Let $\mu_{i,g}$ denote the mean of arm $i$ for the segment $\rho_g$. Let the changepoint gap between the segments $\rho_g$ and $\rho_{g+1}$ be $\Delta_{i,g}^{chg} := |\mu_{i,g} - \mu_{i,g+1}|$. We redefine the optimality gap for the segment $\rho_g$ as $\Delta_{i,g}^{opt} := \mu_{i^*,g} - \mu_{i,g}$. Let $\tau_{c_g}$ denote the first round when the changepoint $t_{c_g}$ is detected and SGR is restarted. Then we define the quantity $N_{0,g}^{chg}$ as the number of times the baseline arm is sampled from rounds $t_{c_g}$ till the detection of changepoint at $\tau_{c_g}$. Finally, we define the delay of detection of the $g$-th changepoint as

$$d_g := \left\lceil K + \left( \max_{i \in [K]} \frac{B(T, \delta)}{(\Delta_{i,g}^{chg})^2} + \frac{B(T, \delta)}{(\Delta_{0,g}^{chg})^2} + N_{0,g}^{bse} \right) 4K \right\rceil \quad (11)$$

such that SGR detects the change at $t_{c_g}$ within $t_{c_g} + 1$ till $t_{c_g} + d_g$ rounds with probability greater than $1 - \delta$. We define the quantity $B(T, \delta) = 16 \log(4 \log_2(T/\delta))$. The quantity $N_{0,g}^{chg}$ denotes the number of samples of the baseline arms after the changepoint $t_{c_g}$ has occurred but not detected and is defined by

$$N_{0,g}^{bse} := \frac{1}{\alpha \mu_{0,g}} \sum_{i \in [K]} \frac{B(T, \delta)}{\max\{\Delta_{i,g}^{opt}, \Delta_{0,g}^{opt} - \Delta_{i,g}^{opt}\}}.$$

Intuitively, $N_{0,g}^{chg}$ is the number of samples required after $t_{c_g}$ has occurred and the safe budget $\widehat{Z}(1 : t)$ falls below 0. In $N_{0,g}^{chg}$ if $\alpha$ is very small, we can still explore other arms as long as the baseline arm $0$ is close to the optimal arm $\mu_{i,g}^*$ (so that $\Delta_{0,g}^{opt}$ is small) while the other arms are clearly sub-optimal (i.e. the $\Delta_{i,g}^{opt}$ are large). If this happens then the sub-optimal arms are quickly discarded, while the $\widehat{Z}(1 : t)$ stays positive and the regret penalty is small. We now define a mild assumption on separation of changepoints which is standard in changepoint detection settings (see Besson and Kaufmann [2019], Besson et al. [2020]). Without this assumption the changepoints can be too frequent and cannot be detected before the next change happens. We require this

assumption for our theoretical guarantees. Note that in the experiments we show that even when this assumption does not hold our proposed algorithms performs well.

**Assumption 1.** *(Separation of changepoints for GCS )* *We assume that the for all $g \in \{0, 1, 2, \ldots, G_T\}$ two consecutive changepoints $t_{c_g}$ and $t_{c_{g+1}}$ are separated as $t_{c_{g+1}} - t_{c_g} \geq 2 \max\{d_g, d_{g+1}\}$, where $d_g$ is stated in (11).*

The Assumption 1 assumes that two consecutive changepoints are separated enough to be detected by the changepoint detector. Note that our detection delay $d_g$ is larger than Besson et al. [2020] because between $t_{c_g} : \tau_{c_g}$ the budget $\widehat{Z}(1 : t)$ may fall below 0 and SGR may need to sample the baseline arm from the next segment $\rho_{g+1}$. We denote an event by $\xi$ and its complement by $\overline{\xi}$. Define the good event $\xi_g^{del}$ that all changepoints $g' \leq g$ have been detected with delay at most $d_{g'}$. Let the safe budget time set $\mathcal{Q}(1 : t) := \left\{ s \in [1 : t] : \widehat{Z}(1 : s) \geq 0 \right\}$ be the set of all rounds $1 : t$ when $\widehat{Z}(1 : s) \geq 0$. We can decompose the expected regret as

$$\sum_{g=1}^{G_T} \left[ \underbrace{\sum_{i=1}^{K} \sum_{s \in \mathcal{Q}(\tau_{c_{g-1}} : t_{c_g} - 1)} \Delta_i^{opt}(s) \mathbb{E}[N_i(s) | \xi_g^{del}(s)] \mathbb{P}\left(\xi_g^{del}(s)\right)}_{\text{Part (A), UCB arm pulled, Safe budget } \widehat{Z}(\tau_{c_{g-1}} : s) \geq 0} \right.$$

$$+ \underbrace{\sum_{s \in \overline{\mathcal{Q}}(\tau_{c_{g-1}} : t_{c_g} - 1)} \Delta_0^{opt}(s) \mathbb{E}[N_0(s) | \xi_g^{del}(s)] \mathbb{P}\left(\xi_g^{del}(s)\right)}_{\text{Part (B), Baseline arm pulled, Safe budget } \widehat{Z}(\tau_{c_{g-1}} : s) < 0}$$

$$+ \underbrace{\sum_{i=1}^{K} \sum_{s \in \mathcal{Q}(t_{c_g} : \tau_{c_g} - 1)} \Delta_i^{opt}(s) \mathbb{E}[N_i(s) | \xi_g^{del}(s)] \mathbb{P}\left(\xi_g^{del}(s)\right)}_{\text{Part (C), Changepoint Pulls, Safe budget } \widehat{Z}(\tau_{c_{g-1}} : s) \geq 0}$$

$$+ \underbrace{\sum_{s \in \overline{\mathcal{Q}}(t_{c_g} : \tau_{c_g} - 1)} \Delta_0^{opt}(s) \mathbb{E}[N_0(s) | \xi_g^{del}(s)] \mathbb{P}\left(\xi_g^{del}(s)\right)}_{\text{Part (D), Changepoint Baseline Pulls, Safe budget } \widehat{Z}(\tau_{c_{g-1}} : s) < 0}$$

$$\left. + \underbrace{\sum_{s=\tau_{c_{g-1}}}^{T} \mathbb{P}(\overline{\xi_g^{del}(s)})}_{\text{Part (E), Total Detection Delay Error}} \right], \quad (12)$$

which follows by dividing the total rounds till $T$ into $G_T$ segments when the changepoint $t_{c_g}$ is detected at $\tau_{c_g}$. We then further subdivide it into two parts $\tau_{c_{g-1}} : t_{c_g} - 1$ (rounds before $t_{c_g}$) and $t_{c_g} : \tau_{c_g} - 1$ (rounds before detection of $t_{c_g}$. The four parts (A)-(D) further divides the two time segments $\tau_{c_{g-1}} : t_{c_g} - 1$ and $t_{c_g} : \tau_{c_g} - 1$ based on the available safe budget and using the definition of $\mathcal{Q}(1 : t)$. Now using Assumption 1 we can show that two consecutive changepoints are separated enough to correctly control the pulls of the baseline arm, and detect the optimal arm given that $\xi_g^{del}$ holds. The main difference

from previous changepoint detection works like Besson and Kaufmann [2019], Mukherjee and Maillard [2019] lies in controlling the detection delay and false alarm under the safety budget constraint. Using the Assumption 1 and changepoint detection Lemma 3 we can show that the detection delay is bounded by $2 \max\{d_g, d_{g+1}\}$ with high probability. We now define a few problem dependent parameters which is key to analyze the regret of SGR. We define the quantity $H_{i,g}^{(1)} := \max\left\{\frac{1}{\Delta_{i,g-1}^{opt}}, \frac{\Delta_{i,g-1}^{opt}}{(\Delta_{i,g-1}^{chg})^2}\right\}$ as the hardness of discarding the sub-optimal arm $i$ and avoiding false detection, the quantity $H_{i,g}^{(2)} := \max_{j \in [K]^+} \frac{\Delta_{i,g}^{opt}}{(\Delta_{j,g}^{chg})^2}$ as the hardness for detecting the changepoint $t_{c_g}$ due to $i$ after the changepoint has happened. Finally, the quantity $H_{i,g}^{(3)} := \frac{\Delta_{\max,g}^{opt}}{\max\{\Delta_{i,g}^{opt}, \Delta_{0,g}^{opt} - \Delta_{i,g}^{opt}\}}$ captures the trade-off of selecting the baseline arm 0 once the changepoint $t_{c_g}$ occurred. The regret of SGR is shown below.

**Theorem 2.** *Let $H_{i,g}^{(1)}, H_{i,g}^{(2)}, H_{i,g}^{(3)}$ is defined above for the segment $\rho_g$. Then the expected regret of SGR is upper bounded by*

$$\mathbb{E}[R_T] \leq O\left(\left(\sum_{g=1}^{G_T}\sum_{i=1}^{K^+}\left(H_{i,g-1}^{(1)} + H_{i,g}^{(2)}\right) + \sum_{g=1}^{G_T}\frac{1}{\alpha\mu_{0,g-1}}\sum_{i=1}^{K}H_{i,g-1}^{(3)}\right.\right.$$
$$\left.\left. + K\sum_{g=1}^{G_T}\frac{1}{\alpha\mu_{0,g}}\sum_{i=1}^{K}H_{i,g}^{(3)}\right)\log\left(\frac{\log_2 T}{\delta}\right)\right). \quad (13)$$

In the result of (13) the first term is the optimality regret suffered before discarding the arm $i$ when the safety budget $\widehat{Z}(1 : t) \geq 0$. The second term denotes the regret suffered for the changepoint detection due to arm $i$. The third term is the regret suffered for the section $\rho_{g-1}$ when the safety budget $\widehat{Z}(1 : t) < 0$. Finally, the fourth term is the regret suffered due to the changepoint $t_{c_g}$ and safety budget $\widehat{Z}(1 : t) < 0$. SGR conducts no forced exploration which results in a fully gap-dependent bound. This result is different than the gap-dependent bound in Mukherjee and Maillard [2019] which does not contain the third and fourth terms in (13). The bound in (13) is more informative than Corollary 1 as it correctly captures the dependence with respect to gaps for each segment $\rho_g$.

# 4 LOCAL CHANGEPOINT DETECTION

In the Local Changepoint Setting (LCS) at any changepoint at least one arm has a change of mean. Recall $G_T$ from (2). We then define the local changepoints $t_{c_0} < t_{c_1} < \ldots < t_{c_{G_T}}$ such that the $g$-th local changepoint is defined as

$$t_{c_g} := \inf\{s > t_{c_{g-1}} : \exists i \in [K], \mu_i(s-1) \neq \mu_i(s)\}.$$

So at a local changepoint the mean of one or more arms may change simultaneously. Let $G_T^i :=$

$\#\{1 \leq s \leq T \mid \mu_i(s-1) \neq \mu_i(s)\}$ denote the number of changepoints only for the $i$-th arm. It follows that $G_T^i \leq G_T$ but for some arms there could be arbitrary difference between these two quantities. Note that $J_T := \sum_{i=1}^{K} G_T^i \leq KG_T$. Define $t_{c_g}^i := \inf\{s > t_{c_{g-1}}^i : \mu_i(s-1) \neq \mu_i(s)\}$ as the $g$-th changepoint for the $i$-th arm. Again $t_{c_0}^i = 1$ for all arms $i \in [K]^+$ by convention. We denote the segment between rounds $t_{c_g}^i$ and $t_{c_{g+1}}^i - 1$ as $\rho_g^i$.

Consider the scenario that a learner has figured out the best arm $i$ in a segment $\rho_g^i$ but then at a local changepoint $t_{c_g}^i$, an arm $j \neq i$ becomes the new optimal arm (but arm $i$ does not change). So it will not be able to detect $t_{c_g}^j$ and will continue sampling arm $i$. Hence the leaner need to conduct forced exploration of all arms to have a good estimate of all arms. This idea is shown in Algorithm 3 where at every round SLR first checks that the safety budget is positive and then either conducts forced exploration of arms with exploration parameters $\gamma$ (to be defined later) or samples the UCB arm $u_t$. If the safety budget is negative then SLR samples the baseline arm so that the budget becomes positive and SLR can explore again. Finally SLR calls the CPD sub-routine with type as "local" to detect local changepoints for an arm. Once the changepoint is detected it restarts only that arm as this is the local changepoint setting. The crucial thing to note is that we conduct forced exploration only when safety budget is available (positive).

---

**Algorithm 3** Safe Local Restart (SLR)

---

1: **Input:** Risk parameter $\alpha$, exploration factor $\gamma$
2: Set $r_i = 1, \forall i \in [K^+]$. Pull each arm once.
3: **for** $t = K^+ + 1, K^+ + 2, \ldots$ **do**
4:     **if** $\widehat{Z}(t) \geq 0$ **then**
5:         **if** $t \bmod \lfloor\frac{K}{\gamma}\rfloor \notin [K]$ **then**
6:             Set $I_t = u_t$ from eq. (6)   ▷*Pull UCB arm*
7:         **else if** $t \bmod \lfloor\frac{K}{\gamma}\rfloor \in [K]$ **then**
8:             Set $I_t = t \bmod \lfloor\frac{K}{\gamma}\rfloor$ ▷*Forced Exploration*
9:     **else if** $\widehat{Z}(t) < 0$ **then**
10:        Set $I_t = 0$             ▷*Baseline arm*
11:     Pull $I_t$ and observe $X_{I_t}(t)$.
12:     Update $\widehat{\mu}_{I_t}(r_{I_t} : t), N_{I_t}(r_{I_t} : t),$ and $\widehat{Z}(r_{I_t} : t)$ in eq. (7).
13:     Call CPD $(\boldsymbol{r}, t, \text{local})$       ▷*Call CPD*

---

## 4.1 REGRET ANALYSIS FOR SLR

We can extend the analysis of SGR to also bound the regret for SLR. The key difference between the two analysis is that SLR needs to bound the regret for each segment $\rho_g^i$ for all arms $i \in K^+$. To this effect we first redefine changepoint gap for any arm $i$ between the segments $\rho_g^i$ and $\rho_{g+1}^i$ as $\Delta_{i,g}^{chg} := |\mu_{i,g} - \mu_{i,g+1}|$, and the optimality gap as $\Delta_{i,g}^{opt} := \mu_{i^*,g} - \mu_{i,g}$. Let $\tau_{c_g}^i$ denote the first round when

the changepoint $t^i_{c_g}$ is detected and SLR is restarted for the arm $i$. Then we define detection delay for the changepoint at $t^i_{c_g}$ as

$$d_{i,g} := \left\lceil \frac{K}{\gamma} + \frac{4}{\gamma}\left(\frac{B(T,\delta)}{(\Delta^{chg}_{i,g})^2} + \frac{B(T,\delta)}{(\Delta^{chg}_{0,g})^2} + N^{bse}_{0,g}\right)\right\rceil. \quad (14)$$

such that $t^i_{c_g}$ is detected within $t^i_{c_g} + 1 : t^i_{c_g} + d_{i,g}$ rounds and $\gamma$ is the exploration rate of SLR . Again we denote $B(T,\delta) = 16\log(4\log_2(T/\delta))$. Note that the delay $d_{i,g}$ scales with the exploration rate $\gamma$ so that SLR while conducting forced exploration can detect $t^i_{c_g}$. Similar assumption has also been taken in Besson and Kaufmann [2019], Besson et al. [2020]. We then define the following assumption for the separation of changepoints between $t^i_{c_g}$ and $t^i_{c_{g+1}}$. Again note that this assumption is only required for theoretical guarantees. Empirically we show that SLR performs well even when the Assumption 2 is violated.

**Assumption 2.** *(Separation of changepoints for LCS )* *We assume that the for all $g \in \{0,1,2,\ldots,G^i_T\}$ two consecutive changepoints $t^i_{c_g}$ and $t^i_{c_{g+1}}$ are separated as $t^i_{c_{g+1}} - t^i_{c_g} \geq 2\max\{d_{i,g}, d_{i,g+1}\}$, where $d_{i,g}$ is defined in (14).*

Next we introduce the quantity $\overline{H^{(2)}_{i,g}} := \frac{\Delta^{opt}_{i,g}}{(\Delta^{chg}_{i,g})^2}$ as the hardness for detecting the $g$-th changepoint for the arm $i$. Note that in the LCS setting the SLR algorithm is restarted only for the arm $i$ and so in the hardness we do not see the max over all arms like the SGR setting. Finally using the Assumption 2 and the same analysis as in Theorem 2 but for each segment $\rho^i_g$ and each arm $i \in [K^+]$ we bound the regret for SLR in Theorem 3.

**Theorem 3.** *Let $H^{(1)}_{i,g}, \overline{H^{(2)}_{i,g}}, H^{(3)}_{i,g}$ is defined above for the segment $\rho_g$. Then the expected regret of SLR is bounded by*

$$\mathbb{E}[R_T] \leq O\Bigg(\Bigg(\sum_{i=1}^{K^+}\sum_{g=1}^{G^i_T}\Big(H^{(1)}_{i,g-1} + \overline{H^{(2)}_{i,g}}\Big) + \sum_{i=1}^{K}\sum_{g=1}^{G^i_T}\frac{1}{\alpha\mu_{0,g-1}}\sum_{i=1}^{K}H^{(3)}_{i,g-1}$$
$$+ K\sum_{g=1}^{G_T}\frac{1}{\alpha\mu_{0,g}}\sum_{i=1}^{K}H^{(3)}_{i,g}\Bigg)\log\left(\frac{\log_2 T}{\delta}\right)\Bigg) + \gamma T. \quad (15)$$

The result in (15) has a similar interpretation to (13) (but with respect to each arm segment $\rho^i_g$ instead of global segment $\rho_g$) except the gap-independent term of $\gamma T$ which results from the forced exploration of arms. We state the following corollary to summarize the result of SGR and SLR in the "easy" case when all the gaps are same.

**Corollary 1.** *(Gap independent bound) Setting $\Delta^{opt}_{i,g} = \Delta^{chg}_{i,g} = \sqrt{\frac{K\log T}{T}}$ for all $i \in [K]^+$ and exploration rate $\gamma = \sqrt{\frac{\log T}{T}}$ we obtain the gap independent regret upper*

bound of SGR and SLR as

$$\mathbb{E}[R_T] \leq O\left(G_T K\sqrt{KT\log T} + \frac{G_T\log T}{\alpha\mu_{0,\min}}\right), \textbf{(SGR )}$$

$$\mathbb{E}[R_T] \leq O\left(G_T\sqrt{KT\log T} + \frac{G_T\log T}{\alpha\mu_{0,\min}}\right), \textbf{(SLR)}$$

*where $\alpha$ is the risk parameter.*

Comparing the above result with GLR-UCB (see Proposition 4) we see that SGR (or SLR ) picks up an additional factor of $1/(\mu_{0,min}\alpha)$ per changepoint which signifies the hardness of finding the safe set of actions for maintaining the safety constraint (1). Further note that SGR suffers an extra factor of $K$ in its bound compared to SLR . This is because in the GCS setting the algorithm restarts by erasing the history of interactions for all arms. Hence, our result mirrors a similar observation in Besson et al. [2020]. Moreover as $\alpha \to 0$ (risky setting) the regret increases proportionally. This is similar to the gap-independent bound in Wu et al. [2016] shown in Proposition 2 which holds for the stochastic setting without any changepoints. The key takeaway from this result is that the piecewise i.i.d. setting under safety constraints is no harder than the conservative stochastic setting of Wu et al. [2016] and piecewise i.i.d. setting given the changepoints are sufficiently separated. Finally we state the lower bound in the safe GCS setting.

**Theorem 3.** *(Lower Bound) Let $\mathcal{E}, \overline{\mathcal{E}}$ be two bandit environment and there exits a global changepoint at $t_{c_1} = T/2$. Let $\alpha > 0$ be the safety parameter and $\mu_{0,\min}$ be the mean of the minimum safety mean over the changepoint segments. Then the lower bound is given by $\mathbb{E}_{\mathcal{E},\overline{\mathcal{E}}}[R_T] \geq$*
$$\left\{\frac{K}{(16e+8)\alpha\mu_{0,\min}} + \frac{\log T}{\alpha\mu_{0,\min}}, \frac{\sqrt{KT}}{\sqrt{32e+16}} + \frac{\log T}{\alpha\mu_{0,\min}}\right\}.$$

The proof is given in Appendix A.6 and follows from the change of measure argument. Additionally, we use the lower bound results from safe bandit setting of Wu et al. [2016] and changepoint detection setting of Gopalan et al. [2021] to arrive at the final result. Note that both of these works do not take into account the safe GCS setting. Finally, comparing the results of Theorem 3 and Corollary 1 we see that SGR matches the lower bound when $G_T = 1$ except a factor of $O(K\sqrt{\log T})$. Similarly, since GCS is a special case of LCS , we see that SLR also matches the lower bound except a factor of $O(K\sqrt{\log T})$.

## 5 EXPERIMENTS

In this section we test SGR and SLR against safety oblivious actively adaptive algorithms GLR-UCB , UCB-CPD as well as passive algorithm D-UCB , and safety aware algorithms CUCB , and UMOSS . A detailed discussion on the algorithms, hyper-parameter tuning, and time complexity of the algorithms is given in Appendix A.7. One further experiment showing the performance of SGR , SLR under

different values of $\alpha$ is shown in Appendix A.7. All codes are provided in supplementary material.

**Global Changepoint:** In this setting all the arms (including baseline) change at every changepoint. The environment consist of 6 arms (including baseline) and the evolution of means with respect to rounds is shown in Figure 1 (Left). The three global changepoints are at $t = 2000, 4000$ and 6000. We set risk parameter $\alpha = 0.7$. The performance of all the algorithms is shown in Figure 2 (Left). The adaptive algorithms like UCB-CPD , GLR-UCB perform well as they detecting the changepoints and restart but they do not satisfy the safety constraints. Note that SGR performs similar to GLR-UCB , UCB-CPD as it also detects the changepoints and restarts as well as satisfy the safety constraint. It outperforms passive algorithm D-UCB , and safety aware algorithm CUCB . The safety aware algorithm CUCB is not suited for the safety constraint (1) under piecewise i.i.d. setting as it always chooses the baseline arm and fail to achieve sub-linear regret.

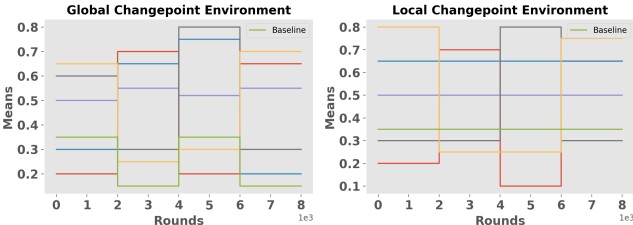

Figure 1: (Left) Global changepoint environment with $T = 8000$, $K^+ = 6$ and changepoints at $t = 2000, 4000$ and 6000. (Middle) Local changepoint environment with $T = 8000$, $K^+ = 6$ and changepoints at $t = 2000, 4000$ and 6000. Note that some arms do not change at these changepoints.

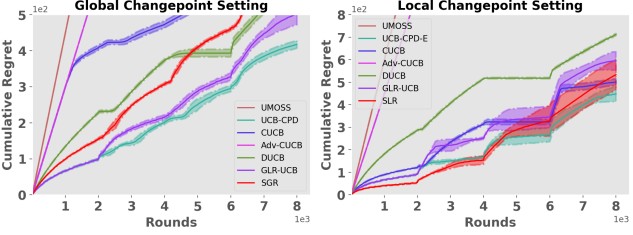

Figure 2: (Left) GCS setting with 3 changepoints and 6 arms. (Right) LCS setting with 3 changepoints and 6 arms.

**Local Changepoint:** In this setting at least one arm changes at every changepoint. We show that the environment in Figure 1 (Right) and the performance of all the algorithms in Figure 2 (Right). The three local changepoints are at $t = 2000, 4000$ and 6000. We set a constant baseline $\mu_0 = 0.35$, and risk parameter $\alpha = 0.7$. Again we see that the safety aware algorithm CUCB fail to achieve sub-linear regret as it always chooses the baseline arm. On the other hand adaptive algorithms like UCB-CPDE , GLR-UCB performs well in

detecting the changepoints but they do not satisfy the safety constraints. Note that SGR performs similar to GLR-UCB , UCB-CPD as it also detects the changepoints and restarts as well as satisfy the safety budget. It outperforms passive algorithm D-UCB , and safety aware algorithm CUCB .

**Real Setting:** We show a real world experiment on the Movielens Dataset. In this experiment none of our modeling assumptions hold. We experiment with the Movielens dataset from February 2003 [Harper and Konstan, 2016], where there are 6k users who give 1M ratings to 4k movies. We obtain a rank-4 approximation of the dataset over 128 users and 128 movies such that all users prefer either movies 7, 13, 16, or 20 (4 user groups). The movies are the arms and we choose 30 movies that have been rated by all the users. Hence, this testbed consists of 30 arms and is run over $T = 8000$. The changepoints are at $t = 2000$, $t = 4000$, and $t = 6000$. Note that at each changepoint the means of some arms may or may not change so this is LCS . For every changepoint segment, we uniform randomly sample an user from different user groups to simulate the piecewise i.i.d environment such that there is a change in the optimal arm. In this environment each arm has has a Gaussian distribution associated with it, where its mean evolve as shown in Figure 4 (Left). The baseline arm is set as $0.35$. As shown in Figure 4 (Right), in this environment SLR outperforms all the other algorithms including CUCB and Adv-CUCB . This is because the means of the arms are close to each other and the baseline arm mean is close to them.

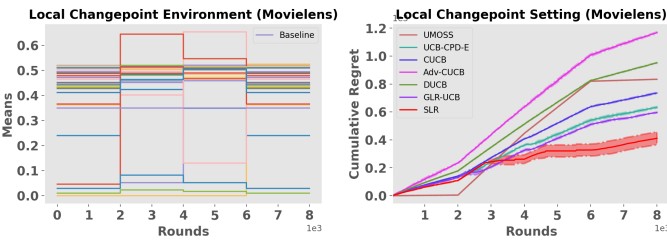

Figure 3: (Left) LCS setting with 3 changepoints and 30 arms. (Right) Regret in Movielens dataset.

# 6 CONCLUSION AND FUTURE WORKS

In this paper we studied the safety aware piecewise i.i.d. bandits under a new safety constraint. We proposed two actively adaptive algorithms SGR and SLR which satisfy the safety constraints as well as detect changepoints and restart. We provided regret bounds on our algorithms and showed how the bounds compare with respect to safety aware bandits as well as adaptive algorithms. We also provided the first matching lower bounds for this setting. Future works include extending our setting to the rested and sleeping bandit setting under safety constraints. We also intend to explore experimental design approaches to piecewise i.i.d settings as in Pukelsheim [2006], Mason et al. [2021], Mukherjee et al. [2022]. Finally, incorporating variance aware techniques [Audibert et al., 2009, Mukherjee et al., 2018] may further improve the the performance of our proposed algorithms.

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

## A APPENDIX

### A.1 PROBABILITY TOOLS, PREVIOUS THEORETICAL RESULTS AND GLRT DISCUSSION

**Proposition 1.** *(Restatement of Theorem 9.2 in Lattimore and Szepesvári [2020])* *Let $X_1, X_2, \ldots, X_T$ be a sequence of independent $\sigma$-subgaussian random variables with $\mathbb{E}[X_1] = \mathbb{E}[X_2] = \ldots = \mathbb{E}[X_T] = 0$ and $M_t = \sum_{s=1}^{t} X_s$. Then for any $\varepsilon > 0$,*

$$\mathbb{P}\left(exists\ t \leq T : M_t \geq \varepsilon\right) \leq \exp\left(-\frac{\varepsilon^2}{2T\sigma^2}\right).$$

**Proposition 2.** *(Restatement of Theorem 2 in Wu et al. [2016])* *In any stochastic environment where the mean of the arms $\mu_i \in [0,1]$ with 1-subgaussian noise, then Conservative UCB (CUCB) satisfies the following with probability at least $1 - \delta$*

$$\sum_{s=1}^{t} \mu_{I_s} \geq (1-\alpha)\mu_0 t \quad for\ all\ t \in \{1, \ldots, T\}, \quad \textbf{(Safety Constraint)}$$

$$\mathbb{E}[R_T] \leq O\left(\sum_{i=1}^{K} \frac{\log T}{\Delta_i^{opt}} + \frac{1}{\alpha\mu_0}\sum_{i=1}^{K} \frac{\log T}{\max\{\Delta_i^{opt}, \Delta_0^{opt} - \Delta_i^{opt}\}}\right), \quad \textbf{(Gap-dependent bound)}$$

$$\mathbb{E}[R_T] \leq O\left(\sqrt{TK\log T} + \frac{K\log T}{\alpha\mu_0}\right). \quad \textbf{(Gap-independent bound)}$$

**Proposition 3.** *(Restatement of Theorem 9 in Wu et al. [2016])* *Let $\mu_i \in [0,1]$ for all $i \in [K]$ and $\mu_0$ satisfies the following*

$$\min\{\mu_0, 1 - \mu_0\} \geq \max\{1/2\sqrt{\alpha}, \sqrt{e + 1/2}\}\sqrt{K/T}.$$

*Then any algorithm satisfies the safety constraint $\mathbb{E}_\mu \sum_{s=1}^{T} X_{s,I_s} \geq (1-\alpha)\mu_0 T$. Moreover there is some $\mu \in [0,1]^K$ such that the expected regret of the algorithm satisfies*

$$\mathbb{E}_\mu[R_T] \geq \max\left\{\frac{K}{(16e+8)\alpha\mu_0}, \frac{\sqrt{KT}}{\sqrt{16e+8}}\right\}.$$

**Proposition 4.** *(Restatement of Corollary 6 and Corollary 9 in Besson et al. [2020])* *The gap-independent bound of GLR-UCB when the number of changepoints is unknown, exploration parameter $\gamma = \sqrt{\frac{\log T}{T}}$, and confidence $\delta = \frac{1}{\sqrt{T}}$ is given by*

$$\mathbb{E}[R_T] \leq O\left(KG_T\sqrt{KT\log T}\right), \quad \textbf{(Global Setting)}$$

$$\mathbb{E}[R_T] \leq O\left(G_T\sqrt{KT\log T}\right). \quad \textbf{(Local Setting)}$$

**Proposition 5.** *(Restatement of Theorem 2 in Gopalan et al. [2021])* *Let $0 < \delta \leq \frac{1}{10}$ and $m \geq 1$ be a priori fixed time such that the probability of an admissible change detector stopping before time $m$ under the null hypothesis (no change) is at most $\delta \in (0,1)$. Let $\tau$ be the stopping time for an algorithm. Let $\mathcal{E} \in \mathbb{R}^K$ and $\overline{\mathcal{E}} \in \mathbb{R}^K$ be two bandit environments consisting of $K$ arms. Define $\mathcal{E}^i$ as the $i$-th component of $\mathcal{E}$. Let $\mathbb{P}_\mathcal{E}$ and $\mathbb{P}_{\overline{\mathcal{E}}}$ be two probability measures induced by some $T$-round interaction of the algorithm with $\mathcal{E}$ and $\overline{\mathcal{E}}$ respectively. Then for any bandit changepoint algorithm satisfying $\mathbb{P}[\tau < m] \leq \delta$, we have*

$$\mathbb{E}[\tau] \geq \min\left\{\frac{\frac{1}{20}\log\frac{1}{\delta}}{\max_{i \in [K]} \text{KL}\left(\mathbb{P}_{\mathcal{E}^i} \| \mathbb{P}_{\overline{\mathcal{E}}^i}\right)}, \frac{m}{2}\right\}.$$

### A.2 PROOF OF REGRET BOUND FOR SAFETY AWARE GLOBAL RESTART

**Theorem 2.** *(Restatement)* *Let $H_{i,g}^{(1)}, H_{i,g}^{(2)}, H_{i,g}^{(3)}$ is defined above for the segment $\rho_g$. Then the expected regret of SGR is upper bounded by*

$$\mathbb{E}[R_T] \leq O\left(\left(\sum_{g=1}^{G_T}\sum_{i=1}^{K^+}\left(H_{i,g-1}^{(1)} + H_{i,g}^{(2)}\right) + \sum_{g=1}^{G_T} \frac{1}{\alpha\mu_{0,g-1}}\sum_{i=1}^{K} H_{i,g-1}^{(3)} + K\sum_{g=1}^{G_T}\frac{1}{\alpha\mu_{0,g}}\sum_{i=1}^{K} H_{i,g}^{(3)}\right)\log\left(\frac{\log_2 T}{\delta}\right)\right). \quad (16)$$

*Proof.* **Step 1 (Regret Decomposition) :** First recall that the safe budget time set $\mathcal{Q}(1:t) := \left\{ s \in [1:t] : \widehat{Z}(1:s) \geq 0 \right\}$ is the set of all rounds $1:t$ when $\widehat{Z}(1:s) \geq 0$. Also recall that the first round the global changepoint $t_{c_g}$ is detected is denoted by $\tau_{c_g}$ defined as follows:

$$\tau_{c_g} := \inf\{t \in \mathbb{N} : \exists s \in [1,t], \exists i \in [K], |\widehat{\mu}_i(1:s) - \widehat{\mu}_i(s+1:t)| > \beta_i(1:s,\delta) + \beta_i(s+1:t,\delta)\}.$$

Let $\xi_g^{del}(s)$ (to be defined later) denote the good event that all changepoints $g' \leq g$ has been successfully detected before the round $s$. We then define the expected regret till round $T$ as follows:

$$\mathbb{E}[R_T] = \sum_{s=1}^{T} \left( \mu_{i^*}(s) - \mathbb{E}[X_{I_s}(s)] \right)$$

$$\overset{(a)}{\leq} \sum_{g=1}^{G_T} \left[ \sum_{s=\tau_{c_{g-1}}}^{t_{c_g}-1} \left( \mu_{i^*}(s) - \mathbb{E}[\mu_{I_s}(s)|\xi_g^{del}(s)] \right) \mathbb{P}\left( \xi_g^{del}(s) \right) + \sum_{s=t_{c_g}}^{\tau_{c_g}-1} \left( \mu_{i^*}(s) - \mathbb{E}[\mu_{I_s}(s)|\xi_g^{del}(s)] \right) \mathbb{P}\left( \xi_g^{del}(s) \right) + \sum_{s=\tau_{c_{g-1}}}^{T} \mathbb{P}(\overline{\xi_g^{del}(s)}) \right]$$

$$\overset{(b)}{=} \sum_{g=1}^{G_T} \left[ \sum_{s\in\mathcal{Q}(\tau_{c_{g-1}}:t_{c_g}-1)} \left( \mu_{i^*}(s) - \mathbb{E}[\mu_{I_s}(s)|\xi_g^{del}(s)] \right) \mathbb{P}\left( \xi_g^{del}(s) \right) + \sum_{s\in\overline{\mathcal{Q}}(\tau_{c_{g-1}}:t_{c_g}-1)} \left( \mu_{i^*}(s) - \mathbb{E}[\mu_{I_s}(s)|\xi_g^{del}(s)] \right) \mathbb{P}\left( \xi_g^{del}(s) \right) \right.$$

$$\left. + \sum_{s=t_{c_g}^k}^{\tau_{c_g}-1} \left( \mu_{i^*}(s) - \mathbb{E}[\mu_{I_s}(s)|\xi_g^{del}(s)] \right) \mathbb{P}\left( \xi_g^{del}(s) \right) ] + \sum_{s=\tau_{c_{g-1}}}^{T} \mathbb{P}(\overline{\xi_g^{del}(s)}) \right]$$

$$\overset{(c)}{=} \sum_{g=1}^{G_T} \left[ \underbrace{\sum_{i=1}^{K} \sum_{s\in\mathcal{Q}(\tau_{c_{g-1}}:t_{c_g}-1)} \Delta_i^{opt}(s)\mathbb{E}[N_i(s)|\xi_g^{del}(s)]\mathbb{P}\left( \xi_g^{del}(s) \right)}_{\textbf{Part (A), UCB arm pulled, Safe budget } \widehat{Z}(\tau_{c_{g-1}}:s) \geq 0} + \underbrace{\sum_{s\in\overline{\mathcal{Q}}(\tau_{c_{g-1}}:t_{c_g}-1)} \Delta_0^{opt}(s)\mathbb{E}[N_0(s)|\xi_g^{del}(s)]\mathbb{P}\left( \xi_g^{del}(s) \right)}_{\textbf{Part (B), Baseline arm pulled, Safe budget } \widehat{Z}(\tau_{c_{g-1}}:s) < 0} \right.$$

$$+ \underbrace{\sum_{i=1}^{K} \sum_{s\in\mathcal{Q}(t_{c_g}:\tau_{c_g}-1)} \Delta_i^{opt}(s)\mathbb{E}[N_i(s)|\xi_g^{del}(s)]\mathbb{P}\left( \xi_g^{del}(s) \right)}_{\textbf{Part (C), Changepoint Pulls, Safe budget } \widehat{Z}(\tau_{c_{g-1}}:s) \geq 0} + \underbrace{\sum_{s\in\overline{\mathcal{Q}}(t_{c_g}:\tau_{c_g}-1)} \Delta_0^{opt}(s)\mathbb{E}[N_0(s)|\xi_g^{del}(s)]\mathbb{P}\left( \xi_g^{del}(s) \right)}_{\textbf{Part (D), Changepoint Baseline Pulls, Safe budget } \widehat{Z}(\tau_{c_{g-1}}:s) < 0}$$

$$\left. + \sum_{s=\tau_{c_{g-1}}}^{T} \underbrace{\mathbb{P}(\overline{\xi_g^{del}(s)})}_{\textbf{Part (E), Total Detection Delay Error}} \right], \tag{17}$$

where $(a)$ follows by introducing the good changepoint detection event $\xi_g^{del}(s)$, $(b)$ follows by introducing the safe budget time set $\mathcal{Q}(\tau_{c_{g-1}} : t_{c_g} - 1)$, and $(c)$ follows by taking into account the safe budget time after the changepoint $t_{c_g}$ has occurred.

**Step 2 (Bounding UCB pulls of sub-optimal arm $i$ in part A):** In this step we bound the total number of samples of sub-optimal arm pulled by using the maximum UCB index $u_t$.

$$\mathbb{E}\left[ \sum_{s\in\mathcal{Q}(\tau_{c_{g-1}}:t_{c_g}-1)} N_i(s) \bigg| \xi_g^{del}(s) \right] = \mathbb{E}\left[ \sum_{s\in\mathcal{Q}(\tau_{c_{g-1}}:t_{c_g}-1)} \mathbf{1}\{I_s = i, N_i(s) \leq \max\{N_{i,g-1}^{opt}, N_{i,g-1}^{chg}\}, \tau_g^{chg} \in [t_{c_g}+1, t_{c_g}+d_g]\} \right]$$

$$+ \mathbb{E}\left[ \sum_{s\in\mathcal{Q}(\tau_{c_{g-1}}:t_{c_g}-1)} \mathbf{1}\{I_s = i, N_i(s) > \max\{N_{i,g-1}^{opt}, N_{i,g-1}^{chg}\}, \tau_g^{chg} \in [t_{c_g}+1, t_{c_g}+d_g]\} \right]$$

$$\overset{(a)}{\leq} \max\{N_{i,g-1}^{opt}, N_{i,g-1}^{chg}\} + d_g + \mathbb{P}\left( \overline{\xi_{i,g-1}^{opt}(s)} \bigcap \overline{\xi_g^{del}(s)} \right)$$

$$\overset{(b)}{\leq} \max\{N_{i,g-1}^{opt}, N_{i,g-1}^{chg}\} + d_g + \mathbb{P}\left( \overline{\xi_{i,g-1}^{opt}(s)} \right).$$

where, in $(a)$ the $N_{i,g-1}^{opt}$ is the maximum number of samples before a sub-optimal arm $i$ is discarded in favor of the optimal arm and $N_{i,g-1}^{chg}$ is maximum number of samples before the changepoint is detected due to arm $i$ (with high probability).

Next the term $d_g$ is the maximum delay for detecting the changepoint $t_{c_g}$ due to some other arm $i$. Note that we have assumed in Assumption 1 that each changepoint $g \in [G_T]$ are separated by $2 \max\{d_g, d_{g+1}\}$. The inequality in $(b)$ follows by dropping the event $\overline{\xi_g^{del}(s)}$.

**Step 3 (Bounding Baseline pulls in part B):** In this step we bound the pulls of the baseline arm when the safety budget $\widehat{Z}(1:t) < 0$. The breakdown of the total pulls follows the same way as in the previous step.

$$
\mathbb{E}\left[\sum_{s \in \overline{\mathcal{Q}}(\tau_{c_{g-1}}:t_{c_g}-1)} N_0(s)|\xi_g^{del}(s)\right] = \mathbb{E}\left[\sum_{s \in \overline{\mathcal{Q}}(\tau_{c_{g-1}}:t_{c_g}-1)} \mathbf{1}\{I_s = 0, N_0(s) \le \max\{N_{0,g-1}^{bse}, N_{0,g-1}^{chg}\}, \tau_g^{chg} \in [t_{c_g}+1, t_{c_g}+d_g]\}\right]
$$

$$
+ \mathbb{E}\left[\sum_{s \in \overline{\mathcal{Q}}(\tau_{c_{g-1}}:t_{c_g}-1)} \mathbf{1}\{I_s = 0, N_0(s) > \max\{N_{0,g-1}^{bse}, N_{0,g-1}^{chg}\}, \tau_g^{chg} \in [t_{c_g}+1, t_{c_g}+d_g]\}\right]
$$

$$
\overset{(a)}{\le} \mathbb{E}\left[\sum_{s \in \overline{\mathcal{Q}}(\tau_{c_{g-1}}:t_{c_g}-1)} \mathbf{1}\{I_s = 0, N_0(s) \le \max\{N_{0,g-1}^{bse}, N_{0,g-1}^{opt} N_{0,g-1}^{chg}\}, \tau_g^{chg} \in [t_{c_g}+1, t_{c_g}+d_g]\}\right]
$$

$$
+ \mathbb{E}\left[\sum_{s \in \overline{\mathcal{Q}}(\tau_{c_{g-1}}:t_{c_g}-1)} \mathbf{1}\{I_s = 0, N_0(s) > \max\{N_{0,g-1}^{bse}, N_{0,g-1}^{opt}, N_{0,g-1}^{chg}\}, \tau_g^{chg} \in [t_{c_g}+1, t_{c_g}+d_g]\}\right]
$$

$$
\overset{(b)}{\le} \max\{N_{0,g-1}^{opt}, N_{0,g-1}^{chg}, N_{0,g-1}^{chg}\} + d_g + \mathbb{P}\left(\overline{\xi_{0,g-1}^{opt}(s)} \bigcap \overline{\xi_g^{del}(s)}\right)
$$

$$
\overset{(c)}{\le} \max\{N_{0,g-1}^{opt}, N_{0,g-1}^{chg}, N_{0,g-1}^{bse}\} + d_g + \mathbb{P}\left(\overline{\xi_{0,g-1}^{opt}(s)}\right) \le \max\{N_{0,g-1}^{opt}, N_{0,g-1}^{chg}\} + N_{0,g-1}^{bse} + d_g + \mathbb{P}\left(\overline{\xi_{0,g-1}^{opt}(s)}\right)
$$

where, in $(a)$ we introduce the optimality pulls of the baseline arm, $(b)$ follows as when the good event $\xi_{0,g-1}^{opt}(s)$ holds then the baseline arm cannot be sampled more than $\max\{N_{0,g-1}^{opt}, N_{0,g-1}^{bse}\}$, and $(c)$ follows by dropping one event from the intersection.

**Step 4 (Bounding Part (C)):** Note that under Assumption 1 the changepoints are separated enough such that the detection delay for the $g$-th changepoint is defined as

$$
d_g := \left\lceil K + 4\left(K \max_{i \in [K]} \frac{B(T, \delta)}{(\Delta_{i,g}^{chg})^2} + \frac{B(T, \delta)}{(\Delta_{0,g}^{chg})^2} + N_{0,g}^{bse}\right)\right\rceil.
$$

where $B(T, \delta) = 16\log(4\log_2(T/\delta))$ and two consecutive global changepoints are separated by $t_{c_{g+1}} - t_{c_g} \ge 2\max\{d_g, d_{g+1}\}$. Then we can bound the total pulls of a sub-optimal arm $i$ in **Part (C)** as follows:

$$
\sum_{s \in \mathcal{Q}(t_{c_g}:\tau_{c_g}-1)} \mathbb{E}\left[N_i(s)|\xi_g^{del}\right] \mathbb{P}\left(\xi_g^{del}\right) \le \mathbb{E}\left[\sum_{s \in \mathcal{Q}(t_{c_g}:\tau_{c_g}-1)} \mathbf{1}\{I_s = i, N_i(s) < N_{i,g}^{chg}, \tau_g^{chg} \in [t_{c_g}+1, t_{c_g}+d_g]\}\right]
$$

$$
+ \mathbb{E}\left[\sum_{s \in \mathcal{Q}(t_{c_g}:\tau_{c_g}-1)} \mathbf{1}\{I_s = i, N_i(s) \ge N_{i,g}^{chg}, \tau_g^{chg} \in [t_{c_g}+1, t_{c_g}+d_g]\}\right]
$$

$$
= \mathbb{E}\left[\sum_{s \in \mathcal{Q}(t_{c_g}:\tau_{c_g}-1)} \mathbf{1}\{I_s = i, N_i(s) < N_{i,g}^{chg}, \tau_{c_g} = \tau_{c_g}^i, \tau_g^{chg} \in [t_{c_g}+1, t_{c_g}+d_g]\}\right]
$$

$$
+ \mathbb{E}\left[\sum_{s \in \mathcal{Q}(t_{c_g}:\tau_{c_g}-1)} \mathbf{1}\{I_s = i, N_i(s) < N_{i,g}^{chg}, \tau_{c_g} > \tau_{c_g}^i, \tau_g^{chg} \in [t_{c_g}+1, t_{c_g}+d_g]\}\right]
$$

$$
+ \mathbb{E}\left[\sum_{s \in \mathcal{Q}(t_{c_g}:\tau_{c_g}-1)} \mathbf{1}\{I_s = i, N_i(s) \ge N_{i,g}^{chg}, \tau_{c_g} = \tau_{c_g}^i, \tau_g^{chg} \in [t_{c_g}+1, t_{c_g}+d_g]\}\right]
$$

$$+ \sum_{i=1}^{K} \mathbb{E}\left[\sum_{s \in \mathcal{Q}(t_{c_g}:\tau_{c_g}-1)} \mathbf{1}\{I_s = i, N_i(s) \geq N_{i,g}^{chg}, \tau_{c_g} > \tau_{c_g}^i, \tau_g^{chg} \in [t_{c_g}+1, t_{c_g}+d_g]\}\right]$$

$$\overset{(a)}{\leq} N_{i,g}^{chg} + \max_{j \in [K]} N_{j,g}^{chg} + \sum_{s \in \mathcal{Q}(t_{c_g}:\tau_{c_g}-1)} \mathbb{P}\left(\xi_{i,g}^{chg}(s)\right) + \sum_{j=1}^{K} \sum_{s \in \mathcal{Q}(t_{c_g}:\tau_{c_g}-1)} \mathbb{P}\left(\xi_{j,g}^{chg}(s)\right) + 4d_g + \sum_{s \in \overline{\mathcal{Q}}(t_{c_g}:\tau_{c_g}-1)} \mathbb{P}(\overline{\xi_g^{del}(s)})$$

$$\leq N_{i,g}^{chg} + \max_{j \in [K]} N_{j,g}^{chg} + \sum_{s=1}^{T} \mathbb{P}\left(\xi_{i,g}^{chg}(s)\right) + \sum_{j=1}^{K} \sum_{s=1}^{T} \mathbb{P}\left(\xi_{j,g}^{chg}(s)\right) + 4d_g + \sum_{s=1}^{T} \mathbb{P}(\overline{\xi_g^{del}(s)})$$

where in $(a)$ the $\max_{j \in [K]} N_{j,g}^{chg}$ is the maximum number of samples required to detect the $g$-th changepoint due to some other arm than $i$.

**Step 5 (Bounding Part (D)):** Again, note that under Assumption 1 the changepoints are separated enough such that the detection delay for the $g$-th changepoint is defined as

$$d_g := \left\lceil K + 4\left(K \max_{i \in [K]} \frac{B(T,\delta)}{(\Delta_{i,g}^{chg})^2} + \frac{B(T\delta)}{(\Delta_{0,g}^{chg})^2} + N_{0,g}^{bse}\right)\right\rceil.$$

where, $B(T,\delta) = 16\log(4\log_2(T/\delta))$ and two consecutive global changepoints are separated by

$$t_{c_{g+1}} - t_{c_g} \geq 2\max\{d_g, d_{g+1}\}.$$

Then we can bound the total pulls of a sub-optimal arm $i$ in **Part (D)** as follows:

$$\sum_{s \in \overline{\mathcal{Q}}(t_{c_g}:\tau_{c_g}-1)} \mathbb{E}\left[N_0(s)|\xi_g^{del}\right] \mathbb{P}\left(\xi_g^{del}\right) \leq \mathbb{E}\left[\sum_{s \in \overline{\mathcal{Q}}(t_{c_g}:\tau_{c_g}-1)} \mathbf{1}\{I_s = 0, N_0(s) < N_{0,g}^{chg}, \tau_g^{chg} \in [t_{c_g}+1, t_{c_g}+d_g]\}\right]$$

$$+ \mathbb{E}\left[\sum_{s \in \overline{\mathcal{Q}}(t_{c_g}:\tau_{c_g}-1)} \mathbf{1}\{I_s = 0, N_0(s) \geq N_{0,g}^{chg}, \tau_g^{chg} \in [t_{c_g}+1, t_{c_g}+d_g]\}\right]$$

$$= \mathbb{E}\left[\sum_{s \in \overline{\mathcal{Q}}(t_{c_g}:\tau_{c_g}-1)} \mathbf{1}\{I_s = 0, N_0(s) < N_{0,g}^{chg}, \tau_{c_g} = \tau_{c_g}^0, \tau_g^{chg} \in [t_{c_g}+1, t_{c_g}+d_g]\}\right]$$

$$+ \mathbb{E}\left[\sum_{s \in \overline{\mathcal{Q}}(t_{c_g}:\tau_{c_g}-1)} \mathbf{1}\{I_s = 0, N_0(s) < N_{0,g}^{chg}, \tau_{c_g} > \tau_{c_g}^0, \tau_g^{chg} \in [t_{c_g}+1, t_{c_g}+d_g]\}\right]$$

$$+ \mathbb{E}\left[\sum_{s \in \overline{\mathcal{Q}}(t_{c_g}:\tau_{c_g}-1)} \mathbf{1}\{I_s = 0, N_0(s) \geq N_{0,g}^{chg}, \tau_{c_g} = \tau_{c_g}^0, \tau_g^{chg} \in [t_{c_g}+1, t_{c_g}+d_g]\}\right]$$

$$+ \sum_{i=1}^{K} \mathbb{E}\left[\sum_{s \in \overline{\mathcal{Q}}(t_{c_g}:\tau_{c_g}-1)} \mathbf{1}\{I_s = 0, N_0(s) \geq N_{i,g}^{chg}, \tau_{c_g} > \tau_{c_g}^0, \tau_g^{chg} \in [t_{c_g}+1, t_{c_g}+d_g]\}\right]$$

$$\leq N_{0,g}^{chg} + \max_{j \in [K]} N_{j,g}^{chg} + \sum_{s \in \overline{\mathcal{Q}}(t_{c_g}:\tau_{c_g}-1)} \mathbb{P}\left(\xi_{0,g}^{chg}(s)\right) + \sum_{j=1}^{K} \sum_{s \in \overline{\mathcal{Q}}(t_{c_g}:\tau_{c_g}-1)} \mathbb{P}\left(\xi_{j,g}^{chg}(s)\right) + 4d_g + \sum_{s \in \overline{\mathcal{Q}}(t_{c_g}:\tau_{c_g}-1)} \mathbb{P}(\overline{\xi_g^{del}(s)})$$

$$\leq N_{0,g}^{chg} + \max_{j \in [K]} N_{j,g}^{chg} + \sum_{s=1}^{T} \mathbb{P}\left(\xi_{0,g}^{chg}(s)\right) + \sum_{j=1}^{K} \sum_{s=1}^{T} \mathbb{P}\left(\xi_{j,g}^{chg}(s)\right) + 4d_g + \sum_{s=1}^{T} \mathbb{P}(\overline{\xi_g^{del}(s)})$$

**Step 6 (Controlling the total delay, Part E):** We first define the total detection delay good event as follows:

$$\xi^{del} := \left\{\forall i \in \{1,\ldots,K\}, \forall g \in \{0,1,2,\ldots,G_T\}, \tau_{c_g}^{chg} \in [t_{c_g}+1, t_{c_g}+d_g]\right\}$$

such that the learner detects all the changepoint $g \leq G_T$ with a detection delay of at most $d_g$ where

$$d_g := \left\lceil K + 4 \left( K \max_{i \in [K]} \frac{\beta_i(1:T,\delta)}{(\Delta_{i,g}^{chg})^2} + \frac{\beta_0(1:T,\delta)}{(\Delta_{0,g}^{chg})^2} + N_{0,g}^{bse} \right) \right\rceil. \tag{18}$$

We define a slightly stronger event than $\xi^{del}$ as follows:

$$\xi_g^{del} := \left\{ \forall g' \leq g, \tau_{c_{g'}}^{chg} \in \left[ t_{c_{g'}} + 1, t_{c_{g'}} + d_{g'} \right] \right\}$$

which signifies that all changepoints before $g$ has been successfully detected. It follows that $\xi^{del} \subseteq \xi_g^{del}$ and $\xi_g^{del}$ is $\mathcal{F}_{\tau_{c_g}^{chg}} - 1$-measurable. Then we can show that

$$
\begin{aligned}
\mathbb{P}(\overline{\xi_g^{del}(s)}) &\leq \sum_{g':g'\leq g} \mathbb{E}\left[ \mathbf{1}\{\overline{\xi_{g'}^{del}(s)}\} \sum_{s=\tau_{c_{g'}}}^{t_{c_{g'}+1}} \mathbf{1}\{\left(\cup_{i\in[K]}\overline{\xi_{i,g'}^{chg}}(\tau_{c_{g'}} : s)\right)\} \mid \mathcal{F}_{\tau_{c_g}} \right] \\
&\overset{(a)}{\leq} \sum_{g':g'\leq g} \mathbf{1}\{\overline{\xi_{g'}^{del}(s)}\} \mathbb{E}\left[ \sum_{s=\tau_{c_{g'}}}^{t_{c_{g'}+1}} \mathbf{1}\{\left(\cup_{i\in[K]}\overline{\xi_{i,g'}^{chg}}(\tau_{c_{g'}} : s)\right)\} \mid \mathcal{F}_{\tau_{c_g}} \right] \\
&\leq \sum_{g':g'\leq g} \mathbf{1}\{\overline{\xi_{g'}^{del}(s)}\} \left[ \sum_{s=\tau_{c_{g'}}}^{t_{c_{g'}+1}} \mathbb{P}\left(\cup_{i\in[K]}\overline{\xi_{i,g'}^{chg}}(\tau_{c_{g'}} : s)\right) \right] \overset{(b)}{\leq} G_T K \sum_{t=1}^{T} \frac{1}{t\ln(t)} \leq G_T K \log(\log(T))
\end{aligned}
$$

where, $(a)$ follows from the definition of $\xi_{i,g}^{chg}(\tau_{c_g} : t_{c_g} : s)$ in Lemma 3, and $(b)$ follows from Lemma 3 and setting $\delta = 1/t$.

**Step 7 (Combining everything):** Combining everything and plugging it in (17) we can show that

$$
\begin{aligned}
\mathbb{E}[R_T] \leq \sum_{g=1}^{G_T} \Bigg[ &\underbrace{\sum_{i=1}^{K} \max\{N_{i,g-1}^{opt}, N_{i,g-1}^{chg}\}\Delta_{i,g-1}^{opt} + d_g\Delta_{i,g-1}^{opt} + \sum_{s=\tau_{c_{g-1}}}^{t_{c_g}-1} \mathbb{P}\left(\overline{\xi_{i,g-1}^{opt}(s)}\right)\Delta_{i,g-1}^{opt}}_{\textbf{Part (A)}} \\
&+ \underbrace{\max\{N_{0,g-1}^{opt}, N_{0,g-1}^{chg}\}\Delta_{0,g-1}^{opt} + N_{0,g-1}^{bse}\Delta_{0,g-1}^{opt} + d_g\Delta_{0,g-1}^{opt} + \sum_{s=\tau_{c_{g-1}}}^{t_{c_g}-1} \mathbb{P}\left(\overline{\xi_{0,g-1}^{opt}(s)}\right)\Delta_{0,g-1}^{opt}}_{\textbf{Part (B)}} \\
&+ \underbrace{\sum_{i=1}^{K}\left[ N_{i,g}^{chg} + \max_{j\in[K]} N_{j,g}^{chg} + 4d_g + \sum_{s=t_{c_g}}^{\tau_{c_g}-1} \mathbb{P}\left(\xi_{i,g}^{chg}(s)\right) + \sum_{j=1}^{K}\sum_{s=t_{c_g}}^{\tau_{c_g}-1} \mathbb{P}\left(\xi_{j,g}^{chg}(s)\right)\right]\Delta_{i,g}^{opt}}_{\textbf{Part (C)}} \\
&+ \underbrace{\left[ N_{0,g}^{chg} + \max_{j\in[K]} N_{j,g}^{chg} + 4d_g + \sum_{s=t_{c_g}}^{\tau_{c_g}-1} \mathbb{P}\left(\xi_{0,g}^{chg}\right) + \sum_{j=1}^{K}\sum_{s=t_{c_g}}^{\tau_{c_g}-1} \mathbb{P}\left(\xi_{j,g}^{chg}(s)\right)\right]\Delta_{0,g}^{opt}}_{\textbf{Part (D)}} \\
&+ \underbrace{\sum_{i=1}^{K}\left( d_g\Delta_{i,g}^{opt} + \sum_{s=t_{c_g}}^{\tau_{c_g}-1} \mathbb{P}(\overline{\xi_g^{del}(s)})\Delta_{i,g}^{opt} \right)}_{\textbf{Part (E)}} \Bigg]
\end{aligned}
$$

$$
\overset{(a)}{\leq} \sum_{g=1}^{G_T} \left[ \sum_{i=1}^{K} \max\left\{ \frac{1}{\Delta_{i,g-1}^{opt}}, \frac{\Delta_{i,g-1}^{opt}}{\left(\Delta_{i,g-1}^{chg}\right)^2} \right\} 8\log\left( \frac{4\log_2(T+1)}{\delta} \right) + d_g K \Delta_{\max,g-1}^{opt} + K\Delta_{\max,g-1}^{opt} \sum_{t=1}^{T} \delta \right.
$$

$$
+ \max\left\{ \frac{1}{\Delta_{0,g-1}^{opt}}, \frac{\Delta_{0,g-1}^{opt}}{\left(\Delta_{0,g-1}^{chg}\right)^2} \right\} 8\log\left( \frac{4\log_2(T+1)}{\delta} \right) + N_{0,g-1}^{bse}\Delta_{\max,g-1}^{opt} + d_g\Delta_{0,g-1}^{opt} + K\Delta_{\max,g-1}^{opt} \sum_{t=1}^{T} \delta
$$

$$
+ \left( \sum_{i=1}^{K} \left( \frac{\Delta_{i,g}^{opt}}{\left(\Delta_{i,g}^{chg}\right)^2} + \max_{j\in[K]} \frac{\Delta_{i,g}^{opt}}{\left(\Delta_{j,g}^{chg}\right)^2} \right) \right) 8\log(\frac{4\log_2(T+1)}{\delta}) + \left( \frac{\Delta_{0,g}^{opt}}{\left(\Delta_{0,g}^{chg}\right)^2} + K\max_{j\in[K]} \frac{\Delta_{j,g}^{opt}}{\left(\Delta_{j,g}^{chg}\right)^2} \right) 8\log(\frac{4\log_2(t+1)}{\delta})
$$

$$
\left. + 8d_g\Delta_{\max,g}^{opt} + 2\left( K\sum_{t=1}^{T}\delta + K^2\sum_{t=1}^{T}\delta + d_g + 6G_T K\log(\log(T)) \right)\Delta_{\max,g}^{opt} \right]
$$

$$
\leq O\Bigg( \sum_{g=1}^{G_T} \left( \sum_{i=1}^{K^+} \max\left\{ \frac{1}{\Delta_{i,g-1}^{opt}}, \frac{\Delta_{i,g-1}^{opt}}{\left(\Delta_{i,g-1}^{chg}\right)^2} \right\} + \sum_{i=1}^{K^+} \max_{j\in[K]^+} \frac{\Delta_{i,g}^{opt}}{\left(\Delta_{j,g}^{chg}\right)^2} \right) \log\left( \frac{\log_2 T}{\delta} \right)
$$

$$
+ N_{0,g-1}^{bse}\Delta_{\max,g-1}^{opt} + K\sum_{g=1}^{G_T}\Delta_{\max,g}^{opt} d_g + KG_T\Delta_{\max,g}^{opt}\sum_{t=1}^{T}\delta \Bigg)
$$

$$
\overset{(b)}{=} O\Bigg( \sum_{g=1}^{G_T}\sum_{i=1}^{K^+} \left( H_{i,g-1}^{(1)} + H_{i,g}^{(2)} \right) \log\left( \frac{\log_2 T}{\delta} \right) + \sum_{g=1}^{G_T}\Delta_{\max,g-1}^{opt} \frac{1}{\alpha\mu_{0,g-1}} \sum_{i\in[K]} \frac{\log(\log_2 T/\delta)}{\max\{\Delta_{i,g-1}^{opt}, \Delta_{0,g-1}^{opt} - \Delta_{i,g-1}^{opt}\}}
$$

$$
+ \sum_{g=1}^{G_T}\Delta_{\max,g}^{opt} \frac{1}{\alpha\mu_{0,g}} \sum_{i\in[K]} \frac{\log(\log_2 T/\delta)}{\max\{\Delta_{i,g}^{opt}, \Delta_{0,g}^{opt} - \Delta_{i,g}^{opt}\}} + KG_T\Delta_{\max,g}^{opt}\sum_{t=1}^{T}\delta \Bigg)
$$

$$
\overset{(c)}{=} O\Bigg( \sum_{g=1}^{G_T}\sum_{i=1}^{K^+} \left( H_{i,g-1}^{(1)} + H_{i,g}^{(2)} \right) \log\left( \frac{\log_2 T}{\delta} \right) + K\sum_{g=1}^{G_T}\frac{1}{\alpha\mu_{0,g-1}}\sum_{i=1}^{K} H_{i,g-1}^{(3)}\log\left( \frac{\log_2 T}{\delta} \right) + K\sum_{g=1}^{G_T}\frac{1}{\alpha\mu_{0,g}}\sum_{i=1}^{K} H_{i,g}^{(3)}\log\left( \frac{\log_2 T}{\delta} \right)
$$

$$
= O\Bigg( \left( \sum_{g=1}^{G_T}\sum_{i=1}^{K^+} \left( H_{i,g-1}^{(1)} + H_{i,g}^{(2)} \right) + K\sum_{g=1}^{G_T}\frac{1}{\alpha\mu_{0,g-1}}\sum_{i=1}^{K} H_{i,g-1}^{(3)} + K\sum_{g=1}^{G_T}\frac{1}{\alpha\mu_{0,g}}\sum_{i=1}^{K} H_{i,g}^{(3)} \right) \log\left( \frac{\log_2 T}{\delta} \right) \Bigg)
$$

where, $(a)$ follows from combining all the steps above, $(b)$ follows by using the definition of

$$
H_{i,g-1}^{(1)} := \max\left\{ \frac{1}{\Delta_{i,g-1}^{opt}}, \frac{\Delta_{i,g-1}^{opt}}{\left(\Delta_{i,g-1}^{chg}\right)^2} \right\}, \qquad H_{i,g}^{(2)} := \max_{j\in[K]^+} \frac{\Delta_{i,g}^{opt}}{\left(\Delta_{j,g}^{chg}\right)^2},
$$

and substituting the value of $d_g$. Finally, $(c)$ follows by using the definition of

$$
H_{i,g}^{(3)} := \frac{\Delta_{\max,g}^{opt}}{\max\{\Delta_{i,g}^{opt}, \Delta_{0,g}^{opt} - \Delta_{i,g}^{opt}\}}.
$$

and dropping the low probability event term as it is dominated by other terms. The claim of the theorem follows. $\square$

## A.3 PROOF OF REGRET BOUND FOR SAFETY AWARE LOCAL RESTART

**Theorem 3.** *(Restatement) Let $H_{i,g}^{(1)}, \overline{H_{i,g}^{(2)}}, H_{i,g}^{(3)}$ is defined above for the segment $\rho_g$. Then the expected regret of* SLR *is bounded by*

$$
\mathbb{E}[R_T] \leq O\Bigg( \left( \sum_{i=1}^{K^+}\sum_{g=1}^{G_T^i} \left( H_{i,g-1}^{(1)} + \overline{H_{i,g}^{(2)}} \right) + \sum_{i=1}^{K}\sum_{g=1}^{G_T^i}\frac{1}{\alpha\mu_{0,g-1}}\sum_{i=1}^{K} H_{i,g-1}^{(3)} + \sum_{i=1}^{K}\sum_{g=1}^{G_T^i}\frac{1}{\alpha\mu_{0,g}}\sum_{i=1}^{K} H_{i,g}^{(3)} \right) \log\left( \frac{\log_2 T}{\delta} \right) \Bigg) + \gamma T
$$

*Proof.* **Step 1 (Regret Decomposition):** Recall that $G_T^i$ is the total number of changepoints for the arm $i$. We define the expected regret till round $T$ as follows:

$$\mathbb{E}[R_T] = \sum_{s=1}^{T} \left( \mu_{i^*}(s) - \mathbb{E}[X_{I_s}(s)] \right)$$

$$\leq \sum_{g=1}^{G_T^i} \left[ \sum_{s=\tau_{c_{g-1}^i}}^{t_{c_g}^i - 1} \left( \mu_{i^*}(s) - \mathbb{E}[\mu_{I_s}(s)|\xi_{i,g}^{del}(s)] \right) \mathbb{P}(\xi_{i,g}^{del}(s)) + \sum_{s=t_{c_g}^i}^{\tau_{c_g}^i - 1} \left( \mu_{i^*}(s) - \mathbb{E}[\mu_{I_s}(s)|\xi_{i,g}^{del}(s)] \right) \mathbb{P}(\xi_{i,g}^{del}(s)) + \mathbb{P}(\overline{\xi_{i,g}^{del}(s)}) \right] + \gamma T$$

$$\overset{(a)}{=} \sum_{g=1}^{G_T^i} \Bigg[ \sum_{s \in \mathcal{Q}(\tau_{c_{g-1}}^i : t_{c_g}^i - 1)} \left( \mu_{i^*}(s) - \mathbb{E}[\mu_{I_s}(s)|\xi_{i,g}^{del}(s)] \right) \mathbb{P}\left( \xi_{i,g}^{del}(s) \right) + \sum_{s \in \overline{\mathcal{Q}}(\tau_{c_{g-1}}^i : t_{c_g}^i - 1)} \left( \mu_{i^*}(s) - \mathbb{E}[\mu_{I_s}(s)|\xi_{i,g}^{del}(s)] \right) \mathbb{P}\left( \xi_{i,g}^{del}(s) \right)$$

$$+ \sum_{s=t_{c_g}^k}^{\tau_{c_g} - 1} \left( \mu_{i^*}(s) - \mathbb{E}[\mu_{I_s}(s)|\xi_g^{del}(s)] \right) \mathbb{P}\left( \xi_{i,g}^{del}(s) \right) + \mathbb{P}(\overline{\xi_{i,g}^{del}(s)}) \Bigg] + \gamma T$$

$$\overset{(b)}{=} \underbrace{\sum_{i=1}^{K} \sum_{g=1}^{G_T^i} \sum_{s \in \mathcal{Q}(\tau_{c_{g-1}}^i : t_{c_g}^i - 1)} \Delta_i^{opt}(s) \mathbb{E}[N_i(s)|\xi_{i,g}^{del}(s)] \mathbb{P}\left( \xi_{i,g}^{del}(s) \right)}_{\textbf{Part (A), UCB arm pulled, Safe budget } \widehat{Z}(\tau_{c_{g-1}} : s) \geq 0} + \underbrace{\sum_{g=1}^{G_T^0} \sum_{s \in \overline{\mathcal{Q}}(\tau_{c_{g-1}}^0 : t_{c_g}^0 - 1)} \Delta_0^{opt}(s) \mathbb{E}[N_0(s)|\xi_{0,g}^{del}(s)] \mathbb{P}\left( \xi_{i,g}^{del}(s) \right)}_{\textbf{Part (B), Baseline arm pulled, Safe budget } \widehat{Z}(\tau_{c_{g-1}} : s) < 0}$$

$$+ \underbrace{\sum_{i=1}^{K} \sum_{g=1}^{G_T^i} \sum_{s \in \mathcal{Q}(t_{c_g} : \tau_{c_g}^i - 1)} \Delta_i^{opt}(s) \mathbb{E}[N_i(s)|\xi_{i,g}^{del}] \mathbb{P}\left( \xi_{i,g}^{del}(s) \right)}_{\textbf{Part (C), Changepoint Pulls, Safe budget } \widehat{Z}(\tau_{c_{g-1}} : s) \geq 0} + \underbrace{\sum_{g=1}^{G_T^0} \sum_{s \in \overline{\mathcal{Q}}(t_{c_g} : \tau_{c_g}^0 - 1)} \Delta_0^{opt}(s) \mathbb{E}[N_0(s)|\xi_{0,g}^{del}] \mathbb{P}\left( \xi_{0,g}^{del}(s) \right)}_{\textbf{Part (D), Changepoint Baseline Pulls, Safe budget } \widehat{Z}(\tau_{c_{g-1}} : s) < 0}$$

$$+ \underbrace{\sum_{i=1}^{K} \sum_{g=1}^{G_T^i} \mathbb{P}(\overline{\xi_{i,g}^{del}(s)})}_{\textbf{Part (E), Total Detection Delay Error}} + \underbrace{\gamma T}_{\textbf{Part (F), Forced Exploration Pulls}} . \tag{19}$$

**Step 2 (Bounding Part (A) - (D)):** Note that under Assumption 2 the changepoints of each arm $i$ are separated enough such that the detection delay for the $g$-th changepoint is defined as

$$d_{i,g} := \left\lceil \frac{K}{\gamma} + \frac{4}{\gamma} \left( \frac{B(T,\delta)}{(\Delta_{i,g}^{chg})^2} + \frac{B(T,\delta)}{(\Delta_{0,g}^{chg})^2} + N_{0,g}^{bse} \right) \right\rceil. \tag{20}$$

where $B(T,\delta) = 16 \log(4 \log_2(T/\delta))$ and $\gamma$ is the exploration rate of $SLR$ and two consecutive changepoints of $i$ are separated by

$$t_{c_{g+1}}^i - t_{c_g}^i \geq 2 \max\{d_{i,g}, d_{i,g+1}\}.$$

So when the changepoint $t_{c_g}^i$ is detected and the total detection delay is controlled for the arm $i$ (shown in step 4) we can show using the (17) and steps 2-4 that

$$\textbf{Part (A)} \leq \max\{N_{i,g-1}^{opt}, N_{i,g-1}^{chg}\} + d_{i,g} + \mathbb{P}\left( \overline{\xi_{i,g-1}^{opt}(s)} \right)$$

$$\textbf{Part (B)} \leq \max\{N_{0,g-1}^{bse}, N_{0,g-1}^{chg}\} + N_{0,g-1}^{bse} + d_{i,g} + \mathbb{P}\left( \overline{\xi_{0,g-1}^{opt}(s)} \right)$$

$$\textbf{Part (C)} \leq N_{i,g}^{chg} + \sum_{s=1}^{T} \mathbb{P}\left( \overline{\xi_{i,g}^{chg}(s)} \right) + 4d_{i,g} + \mathbb{P}(\overline{\xi_{i,g}^{del}(s)})$$

$$\textbf{Part (D)} \leq N_{0,g}^{chg} + \sum_{s=1}^{T} \mathbb{P}\left( \overline{\xi_{0,g}^{chg}(s)} \right) + 4d_{i,g} + \mathbb{P}(\overline{\xi_{i,g}^{del}(s)})$$

**Step 3 (Controlling the total delay):** We want to control the total detection delay of individual arms $i$. Again we define the total detection delay good event for the $i$-th arm as follows:

$$\xi_i^{del} := \left\{ \forall i \in \{1, \ldots, K\}, \forall g \in \{0, 1, 2, \ldots, G_T^i\}, \tau_{c_g}^i \in \left[ t_{c_g}^i + 1, t_{c_g}^i + d_{i,g} \right] \right\}$$

such that the learner detects all the changepoint $g \le G_T^i$ with a detection delay of at most $d_{i,g}$ where

$$d_{i,g} := \left\lceil \frac{K}{\gamma} + \frac{4}{\gamma} \left( \frac{B(T, \delta)}{(\Delta_{i,g}^{chg})^2} + \frac{B(T, \delta)}{(\Delta_{0,g}^{chg})^2} + N_{0,g}^{bse} \right) \right\rceil. \tag{21}$$

where $B(T, \delta) = 16 \log(4 \log_2(T/\delta))$. We define a slightly stronger event than $\xi_i^{del}$ as follows:

$$\xi_{i,g}^{del} := \left\{ \forall g' \le g, \tau_{c_{g'}}^i \in \left[ t_{c_{g'}}^i + 1, t_{c_{g'}}^i + d_{i,g'} \right] \right\}$$

which signifies that all changepoints before $g$ has been successfully detected. It follows that $\xi_i^{del} \subseteq \xi_{i,g}^{del}$ and $\xi_{i,g}^{del}$ is $\mathcal{F}_{\tau_{c_g}^i} - 1$-measurable. Then we can show that

$$\mathbb{P}(\overline{\xi_{i,g}^{del}}) \le \sum_{g':g' \le g} \mathbb{E} \left[ \mathbf{1}\{\overline{\xi_{i,g'}^{del}(s)}\} \sum_{s=\tau_{c_{g'}}^i}^{t_{c_{g'+1}}^i} \mathbf{1}\{ \left( \cup_{i \in [K]} \overline{\xi_{i,g'}^{chg}}(\tau_{c_{g'}}^i : s) \right) \} \mid \mathcal{F}_{\tau_{c_g}^i} \right]$$

$$\overset{(a)}{\le} \sum_{g':g' \le g} \mathbf{1}\{\overline{\xi_{i,g'}^{del}(s)}\} \mathbb{E} \left[ \sum_{s=\tau_{c_{g'}}^i}^{t_{c_{g'+1}}} \mathbf{1}\{ \left( \cup_{i \in [K]} \overline{\xi_{i,g'}^{chg}}(\tau_{c_{g'}}^i : s) \right) \} \mid \mathcal{F}_{\tau_{c_g}^i} \right]$$

$$\le \sum_{g':g' \le g} \mathbf{1}\{\overline{\xi_{i,g'}^{del}(s)}\} \left[ \sum_{s=\tau_{c_{g'}}^i}^{t_{c_{g'+1}}^i} \mathbb{P}\left( \cup_{i \in [K]} \overline{\xi_{i,g'}^{chg}}(\tau_{c_{g'}}^i : s) \right) \right]$$

$$\overset{(b)}{\le} G_T^i K \sum_{t=1}^{T} \frac{1}{t \ln(t)} \le G_T^i K \log(\log(T))$$

where, $(a)$ follows from the definition of $\xi_{i,g}^{chg}(\tau_{c_g}^i : t_{c_g}^i : s)$ in Lemma 3, and $(b)$ follows from Lemma 3 and setting $\delta = 1/t$.

**Step 4 (Combining everything):** Combining everything and plugging it in (19) we can show that

$$\mathbb{E}[R_T] \le \underbrace{\sum_{i=1}^{K} \sum_{g=1}^{G_T^i} \left[ \max\{N_{i,g-1}^{opt}, N_{i,g-1}^{chg}\} \Delta_{i,g-1}^{opt} + d_{i,g} \Delta_{i,g-1}^{opt} + \sum_{s=\tau_{c_{g-1}}^i}^{t_{c_g}^i - 1} \mathbb{P}\left( \overline{\xi_{i,g-1}^{opt}(s)} \right) \Delta_{i,g-1}^{opt} \right]}_{\textbf{Part (A)}}$$

$$+ \underbrace{\sum_{g=1}^{G_T^0} \left[ \max\{N_{0,g-1}^{opt}, N_{0,g-1}^{chg}\} \Delta_{0,g-1}^{opt} + N_{0,g-1}^{bse} \Delta_{0,g-1}^{opt} + d_{0,g} \Delta_{0,g-1}^{opt} + \sum_{s=\tau_{c_{g-1}}^i}^{t_{c_g}^i - 1} \mathbb{P}\left( \overline{\xi_{0,g-1}^{opt}(s)} \right) \Delta_{0,g-1}^{opt} \right]}_{\textbf{Part (B)}}$$

$$+ \underbrace{\sum_{i=1}^{K} \sum_{g=1}^{G_T^i} \left[ N_{i,g}^{chg} + 4 d_{i,g} + \sum_{s=t_{c_g}^i}^{\tau_{c_g}^i - 1} \mathbb{P}\left( \xi_{i,g}^{chg}(s) \right) \right] \Delta_{i,g}^{opt}}_{\textbf{Part (C)}} + \underbrace{\sum_{g=1}^{G_T^0} \left[ N_{0,g}^{chg} + 4 d_{i,g} + \sum_{s=t_{c_g}^i}^{\tau_{c_g}^i - 1} \mathbb{P}\left( \xi_{0,g}^{chg}(s) \right) \right] \Delta_{0,g}^{opt}}_{\textbf{Part (D)}}$$

$$+ \underbrace{\sum_{i=1}^{K} \sum_{g=1}^{G_T^i} \left( d_{i,g} \Delta_{i,g}^{opt} + \sum_{s=t_{c_g}^i}^{\tau_{c_g}^i - 1} \mathbb{P}(\overline{\xi_{i,g}^{del}(s)}) \Delta_{i,g}^{opt} \right)}_{\textbf{Part (E)}} + \underbrace{\gamma T}_{\textbf{Part (F)}}$$

$$
\overset{(a)}{\leq} \sum_{i=1}^{K} \sum_{g=1}^{G_T^i} \max\left\{\frac{1}{\Delta_{i,g-1}^{opt}}, \frac{\Delta_{i,g-1}^{opt}}{\left(\Delta_{i,g-1}^{chg}\right)^2}\right\} 8\log\left(\frac{4\log_2(t+1)}{\delta}\right) + \sum_{i=1}^{K}\sum_{g=1}^{G_T^i} d_{i,g} K \Delta_{\max,g-1}^{opt} + K G_T \Delta_{\max,g-1}^{opt} \sum_{t=1}^{T}\delta
$$

$$
+ \sum_{g=1}^{G_T^0} \max\left\{\frac{1}{\Delta_{0,g-1}^{opt}}, \frac{\Delta_{0,g-1}^{opt}}{\left(\Delta_{0,g-1}^{chg}\right)^2}\right\} 8\log\left(\frac{4\log_2(t+1)}{\delta}\right) + N_{0,g-1}^{bse}\Delta_{\max,g-1}^{opt} + \sum_{g=1}^{G_T^0} d_{i,g}\Delta_{0,g-1}^{opt} + K\Delta_{\max,g-1}^{opt}\sum_{t=1}^{T}\delta
$$

$$
+ \sum_{i=1}^{K}\sum_{g=1}^{G_T^i} \left(\frac{\Delta_{i,g}^{opt}}{\left(\Delta_{i,g}^{chg}\right)^2}\right) 8\log\left(\frac{4\log_2(t+1)}{\delta}\right) + \sum_{g=1}^{G_T^0}\left(\frac{\Delta_{0,g}^{opt}}{\left(\Delta_{0,g}^{chg}\right)^2}\right) 8\log\left(\frac{4\log_2(t+1)}{\delta}\right)
$$

$$
+ 8\sum_{i=1}^{K}\sum_{g=1}^{G_T^i} d_{i,g}\Delta_{\max,g}^{opt} + 2\left(\sum_{i=1}^{K}\sum_{g=1}^{G_T^i}\sum_{t=1}^{T}\delta + \sum_{i=1}^{K}\sum_{g=1}^{G_T^i} d_{i,g} + 6\sum_{i=1}^{K}\sum_{g=1}^{G_T^i}\log(\log(T))\right)\Delta_{i,g}^{opt} + \gamma T
$$

$$
\leq O\left(\sum_{i=1}^{K^+}\sum_{g=1}^{G_T^i}\left(\max\left\{\frac{1}{\Delta_{i,g-1}^{opt}}, \frac{\Delta_{i,g-1}^{opt}}{\left(\Delta_{i,g-1}^{chg}\right)^2}\right\} + \frac{\Delta_{i,g}^{opt}}{\left(\Delta_{i,g}^{chg}\right)^2}\right)\log\left(\frac{\log_2 T}{\delta}\right) + \sum_{i=1}^{K}\sum_{g=1}^{G_T^i}\Delta_{\max,g}^{opt} d_{i,g}\right.
$$

$$
\left. + N_{0,g-1}^{bse}\Delta_{0,g-1}^{opt} + \sum_{i=1}^{K}\sum_{g=1}^{G_T^i}\Delta_{i,g}^{opt}\sum_{t=1}^{T}\delta\right) + \gamma T
$$

$$
\overset{(b)}{=} O\left(\sum_{i=1}^{K^+}\sum_{g=1}^{G_T^i}\left(H_{i,g-1}^{(1)} + \overline{H_{i,g}^{(2)}}\right)\log\left(\frac{\log_2 T}{\delta}\right) + \sum_{i=1}^{K}\sum_{g=1}^{G_T^i}\Delta_{\max,g}^{opt}\frac{1}{\alpha\mu_{0,g-1}}\sum_{i\in[K]}\frac{\log(\log_2 T/\delta)}{\max\{\Delta_{i,g-1}^{opt}, \Delta_{0,g-1}^{opt} - \Delta_{i,g-1}^{opt}\}}\right.
$$

$$
\left. + \sum_{i=1}^{K}\sum_{g=1}^{G_T^i}\Delta_{\max,g}^{opt}\frac{1}{\alpha\mu_{0,g}}\sum_{i\in[K]}\frac{\log(\log_2 T/\delta)}{\max\{\Delta_{i,g}^{opt}, \Delta_{0,g}^{opt} - \Delta_{i,g}^{opt}\}} + K G_T \Delta_{\max}^{opt}\sum_{t=1}^{T}\delta\right) + \gamma T
$$

$$
\overset{(c)}{=} O\left(\left(\sum_{i=1}^{K^+}\sum_{g=1}^{G_T^i}\left(H_{i,g-1}^{(1)} + \overline{H_{i,g}^{(2)}}\right) + \sum_{i=1}^{K}\sum_{g=1}^{G_T^i}\frac{1}{\alpha\mu_{0,g-1}}\sum_{i=1}^{K}H_{i,g-1}^{(3)} + \sum_{i=1}^{K}\sum_{g=1}^{G_T^i}\frac{1}{\alpha\mu_{0,g}}\sum_{i=1}^{K}H_{i,g}^{(3)}\right)\log\left(\frac{\log_2 T}{\delta}\right)\right) + \gamma T
$$

where, $(a)$ follows from combining all the steps above, $(b)$ follows by using the definition of

$$
H_{i,g-1}^{(1)} := \max\left\{\frac{1}{\Delta_{i,g-1}^{opt}}, \frac{\Delta_{i,g-1}^{opt}}{\left(\Delta_{i,g-1}^{chg}\right)^2}\right\}, \qquad \overline{H_{i,g}^{(2)}} := \frac{\Delta_{i,g}^{opt}}{\left(\Delta_{i,g}^{chg}\right)^2},
$$

and substituting the value of $d_{i,g}$. Finally, $(c)$ follows by using the definition of

$$
H_{i,g}^{(3)} := \frac{\Delta_{\max,g}^{opt}}{\max\{\Delta_{i,g}^{opt}, \Delta_{0,g}^{opt} - \Delta_{i,g}^{opt}\}}.
$$

The claim of the theorem follows. □

## A.4 PROOF OF COROLLARY 1

**Corollary 1.** *(Gap independent bound, Restatement)* Setting $\Delta_{i,g}^{opt} = \Delta_{i,g}^{chg} = \sqrt{\frac{K\log T}{T}}$ *for all* $i \in [K]^+$ *and exploration rate* $\gamma = \sqrt{\frac{\log T}{T}}$ *we obtain the gap independent regret upper bound of* SGR *and* SLR *as*

$$
\mathbb{E}[R_T] \leq O\left(G_T K\sqrt{KT\log T} + \frac{G_T\log T}{\alpha\mu_{0,\min}}\right), \text{(SGR )}
$$

$$
\mathbb{E}[R_T] \leq O\left(G_T\sqrt{KT\log T} + \frac{G_T\log T}{\alpha\mu_{0,\min}}\right), \text{(SLR)}
$$

*where* $\alpha$ *is the risk parameter.*

*Proof.* The proof directly follows from the result of Theorem 2 and Theorem 3. We first recall the result of Theorem 2 below

$$\mathbb{E}[R_t] \leq O\left(\left(\sum_{g=1}^{G_T}\sum_{i=1}^{K^+}\left(H_{i,g-1}^{(1)} + H_{i,g}^{(2)}\right) + K\sum_{g=1}^{G_T}\frac{1}{\alpha\mu_{0,g-1}}\sum_{i=1}^{K}H_{i,g-1}^{(3)} + K\sum_{g=1}^{G_T}\frac{1}{\alpha\mu_{0,g}}\sum_{i=1}^{K}H_{i,g}^{(3)}\right)\log\left(\frac{\log_2 T}{\delta}\right)\right).$$

(22)

Then using the fact that $\Delta_{i,g}^{opt} = \Delta_{i,g}^{chg} = \sqrt{\frac{K\log T}{T}}$ for all $i \in [K]^+$ we get that

$$H_{i,g-1}^{(1)} := \max\left\{\frac{1}{\Delta_{i,g-1}^{opt}}, \frac{\Delta_{i,g-1}^{opt}}{\left(\Delta_{i,g-1}^{chg}\right)^2}\right\} = \sqrt{\frac{T}{K\log T}},$$

$$H_{i,g}^{(2)} := \max_{j\in[K]^+}\frac{\Delta_{i,g}^{opt}}{\left(\Delta_{j,g}^{chg}\right)^2} = \sqrt{\frac{T}{K\log T}},$$

$$H_{i,g}^{(3)} := \frac{\Delta_{\max,g}^{opt}}{\max\{\Delta_{i,g}^{opt}, \Delta_{0,g}^{opt} - \Delta_{i,g}^{opt}\}} = 1.$$

Substituting all of the above back in Equation (22) and setting $\delta = \frac{1}{T}$ we get that

$$\mathbb{E}[R_T] \leq O\left(G_T K\sqrt{KT\log T} + \frac{G_T\log T}{\alpha\mu_{0,\min}}\right).$$

Next we recall the result of Theorem 3 below

$$\mathbb{E}[R_t] \leq O\left(\left(\sum_{i=1}^{K^+}\sum_{g=1}^{G_T^i}\left(H_{i,g-1}^{(1)} + \overline{H_{i,g}^{(2)}}\right) + \sum_{i=1}^{K}\sum_{g=1}^{G_T^i}\frac{1}{\alpha\mu_{0,g-1}}\sum_{i=1}^{K}H_{i,g-1}^{(3)} + \sum_{i=1}^{K}\sum_{g=1}^{G_T^i}\frac{1}{\alpha\mu_{0,g}}\sum_{i=1}^{K}H_{i,g}^{(3)}\right)\log\left(\frac{\log_2 T}{\delta}\right)\right) + \gamma T.$$

(23)

Again using the fact that $\Delta_{i,g}^{opt} = \Delta_{i,g}^{chg} = \sqrt{\frac{K\log T}{T}}$ for all $i \in [K]^+$ it follows that $\overline{H_{i,g}^{(2)}} = \sqrt{T/K\log T}$. Substituting this back in (23) we get that

$$\mathbb{E}[R_T] \leq O\left(G_T\sqrt{KT\log T} + \frac{G_T\log T}{\alpha\mu_{0,\min}}\right).$$

The claim of the corollary follows. □

## A.5  SUPPORT LEMMA

### A.5.1  Concentration of the Optimality Event

**Lemma 1.** *Let $X_i(1), X_i(2), \ldots, X_i(t)$ be $t$ samples observed for arm $i$. Define the optimality good event $\xi_i^{opt}(1:t) :=$
$\{\forall s \in [1:t], |\widehat{\mu}_i(1:s) - \mu_i(1:s)| \leq \beta_i(1:s,\delta)\}$ where $\beta_i(1:s,\delta) := \sqrt{\frac{2\log(4\log_2(t+1)/\delta)}{s}}$. Then the probability
of the event $\xi_i^{opt}(1:t)$ is bounded by*

$$\mathbb{P}\left(\xi_i^{opt}(1:t)\right) \geq 1 - \delta.$$

*Proof.* We will use the peeling argument to get the desired bound. We first restate the bad event of deviation as

$$\overline{\xi_i^{opt}}(1:t) := \left\{\exists s \in [1:t], |\widehat{\mu}_i(1:s) - \mu_i(1:s)| > \sqrt{\frac{2\log(4\log_2(t+1)/\delta)}{s}}\right\}.$$

(24)

Next we bound the one side of the inequality as follows:

$$\mathbb{P}\left(\exists s \geq 1, \widehat{\mu}_i(1:s) - \mu_i(1:s) \geq \sqrt{\frac{2\log(4\log_2(t+1)/\delta)}{s}}\right) = \mathbb{P}\left(\exists s \geq 1, s\underbrace{(\widehat{\mu}_i(1:s) - \mu_i(1:s))}_{:=Y_s} \geq \sqrt{\frac{2s^2\log(4\log_2(t+1)/\delta)}{s}}\right)$$

$$\overset{(a)}{\leq} \sum_{u=0}^{\lfloor \log_2 t \rfloor} \mathbb{P}\left(\exists s \in [2^u, 2^{u+1}], M_s \geq \sqrt{2s\log(4\log_2(t+1)/\delta)}\right)$$

$$\overset{(b)}{\leq} \sum_{u=0}^{\lfloor \log_2 t \rfloor} \mathbb{P}\left(\exists s \leq 2^{u+1}, M_s \geq \sqrt{2.2^u\log(4\log_2(t+1)/\delta)}\right)$$

$$\overset{(c)}{\leq} \sum_{u=0}^{\lfloor \log_2 t \rfloor} \exp\left(-\frac{(\sqrt{2.2^u\log(4\log_2(t+1)/\delta)})^2}{2.2^u\sigma^2}\right)$$

$$\overset{(d)}{\leq} \sum_{u=0}^{\lfloor \log t \rfloor} \exp\left(-\frac{2^u\log(4\log_2(t+1)/\delta)}{2^u}\right)$$

$$\overset{(e)}{\leq} (\log_2(t+1))\exp\left(-\log(4\log_2(t+1)/\delta)\right) \leq \delta/4,$$

$$\tag{25}$$

where, $(a)$ follows by dividing the rounds till $t$ into geometric grid of size $[2^u, 2^{u+1}]$, $s \leq \lceil \log t \rceil$ and $M_s = \sum_{t'=1}^s (X_i(t') - \mathbb{E}[X_i(t')]) = \sum_{t'=1}^s Y_{t'}$. Further note that $\mathbb{E}[Y_{t'}] = 0$, so we can apply the maximal inequality. Then $(b)$ follows by upper bounding $s$ by $2^{u+1}$, $(c)$ follows by applying Proposition 1, $(d)$ follows as $\sigma^2 = 1$, $(e)$ follows as with $s \leq \lceil \log t \rceil$ we can have $(\log t + 1)$ such combinations. Similarly, we can show that,

$$\mathbb{P}\left(\exists s \geq 1, \widehat{\mu}_i(1:s) - \mu_i(1:s) \leq -\sqrt{\frac{2\log(4\log_2(t+1)/\delta)}{s}}\right) \leq \delta/4 \tag{26}$$

Combining the equations (25) and (26) we get that,

$$\mathbb{P}\left(\overline{\xi_i^{opt}}(1:t)\right) \leq \frac{\delta}{2} < \delta.$$

It follows then $\mathbb{P}\left(\xi_i^{opt}(1:t)\right) \geq 1 - \delta$. Hence the claim of the lemma follows. □

### A.5.2 Critical Samples for Optimality Detection

**Lemma 2.** *Define the optimality event $\xi_i^{opt}(s:t)$ as in Lemma 1. Then the expected number of times the sub-optimal arm $i$ is sampled based on the UCB sampling rule from round $1$ till $t$ is given by*

$$\mathbb{E}[N_i(1:t)] \leq N_i^{opt}(1:t) + \sum_{s=N_i^{opt}(1:t)+1}^{t} \mathbb{P}\left(\overline{\xi_i^{opt}}(s:t)\right)$$

*where, $N_i^{opt}(1:t) = \dfrac{8\log(4\log_2(t+1)/\delta)}{(\Delta_i^{opt})^2}$.*

*Proof.* First note that if a sub-optimal arm $i$ is chosen at round $t$ then $U_i(1:s) > U_{i^*}(1:s)$. This is possible under the following three events

$$\{\widehat{\mu}_i(1:s) \geq \mu_i(1:s) + \beta_i(1:s,\delta)\}, \quad \{\widehat{\mu}_{i^*}(1:s) \leq \mu_{i^*}(1:s) - \beta_i(1:s,\delta)\}, \quad \{\mu_{i^*}(1:s) - \mu_i(1:s) < 2\beta_i(1:s,\delta)\}. \tag{27}$$

Now recall from Lemma 1 that at round $s \in [t]$, the event $\xi_i^{opt}(1:t)$ holds with $1 - \delta$ probability. We define the optimal stopping time $\tau_{i,s}^{opt}(1:t)$ as follows:

$$\tau_{i,s}^{opt}(1:t) = \min\{t' \in [1:t] : N_i(1:t') = s\}.$$

It follows immediately that $\tau_{i,s}^{opt}(1:t)$ is $\mathcal{F}_{t-1}$ measurable. Also note that that the third event in (27) is not possible for

$$\mu_{i^*}(1:t) - \mu_i(1:t) < 2\beta_i(1:t,\delta) \implies \Delta_i^{opt} < 2\sqrt{\frac{2\log(4\log_2(t+1)/\delta)}{N_i^{opt}(1:t)}} \implies N_i^{opt}(1:t) > \frac{8\log(4\log_2(t+1)/\delta)}{(\Delta_i^{opt})^2}.$$

Then we can bound the expected number of pulls for any sub-optimal arm $i$ as follows:

$$\mathbb{E}[N_i(1:t)] = \mathbb{E}\Big[\sum_{s=1}^{N_i^{opt}(1:t)} \mathbf{1}\{\tau_{i,s}^{opt}(1:t) \le N_i^{opt}(1:t)\}\Big] + \mathbb{E}\Big[\sum_{s=N_i^{opt}(1:t)+1}^{t} \mathbf{1}\{\tau_{i,s}^{opt}(1:t) > N_i^{opt}(1:t)\}\Big]$$

$$= N_i^{opt}(1:t) + \sum_{s=N_i^{opt}(1:t)+1}^{t} \mathbb{P}\big(\tau_{i,s}^{opt}(1:t) > N_i^{opt}(1:t)\big)$$

$$\le N_i^{opt}(1:t) + \sum_{s=N_i^{opt}(1:t)+1}^{t} \mathbb{P}\big(\{I_s = i, \widehat{\mu}_i(1:s-1) + \beta_i(1:s-1,\delta) > \widehat{\mu}_{i^*}(1:s-1) + \beta_{i^*}(1:s-1,\delta)\}\big)$$

$$\le N_i^{opt}(1:t) + \sum_{s=N_i^{opt}(1:s-1)+1}^{t} \mathbb{P}\bigg(\{\widehat{\mu}_i(1:s-1) \ge \mu_i(1:s-1) + \beta_i(1:s-1,\delta)\}$$

$$\bigcup \{\widehat{\mu}_{i^*}(1:s-1) \le \mu_{i^*}(1:s-1) - \beta_{i^*}(1:s-1,\delta)\}\bigg)$$

$$\le N_i^{opt}(1:t) + \sum_{s=N_i^{opt}(1:t)+1}^{t} \mathbb{P}\big(\overline{\xi_i^{opt}}(s:t)\big) + \sum_{s=N_i^{opt}(1:t)+1}^{t} \mathbb{P}\big(\overline{\xi_{i^*}^{opt}}(s:t)\big).$$

The claim of the lemma follows. □

### A.5.3 Concentration of Changepoint Event

**Lemma 3.** *Let $\widehat{\mu}_i(1:s)$ be the empirical mean of $s$ i.i.d. observations with mean $\mu_i(1:s)$, and $\widehat{\mu}_i(s+1:t)$ be the empirical mean of $(t-s)$ i.i.d. observations with mean $\mu_i(s+1:t)$. Let $t_{c_g}^i$ be the round such that $\mu_i(t_{c_g}^i) \ne \mu_i(t_{c_g}^i + 1)$. Let at round $s$ the policy raises an alarm following the changepoint good event $\xi_{i,g}^{chg}(1:t)$. Define the changepoint good event*

$$\xi_{i,g}^{chg}(1:t) := \{\exists s \in [1:t], |\widehat{\mu}_i(1:s) - \widehat{\mu}_i(s+1:t)| \ge \beta_i(1:s,\delta) + \beta_i(s+1:t,\delta)\}$$

*where $\beta_i(1:s,\delta) := \sqrt{\dfrac{2\log(4\log_2(t+1)/\delta)}{s}}$ and $\beta_i(s+1:t,\delta) := \sqrt{\dfrac{2\log(4\log_2(t+1)/\delta)}{t-s}}$. Then the probability of the event $\xi_i^{chg}(1:t)$ is bounded by*

$$\mathbb{P}\big(\xi_{i,g}^{chg}(1:t)\big) \ge 1 - \delta.$$

*Proof.* Recall that $\widehat{\mu}_i(1:s)$ is the empirical mean of $s$ i.i.d. observations with mean $\mu_i(1:s)$, and $\widehat{\mu}_i(s+1:t)$ is the empirical mean of $(t-s)$ i.i.d. observations with mean $\mu_i(s+1:t)$. We will again use the peeling argument to get the desired bound. We first restate the good event as follows:

$$\xi_{i,g}^{chg}(1:t) := \left\{\exists s \in [1:t], \left|\widehat{\mu}_i(1:s) - \widehat{\mu}_i(s+1:t)\right| > \sqrt{\frac{2\log(4\log_2(t+1)/\delta)}{s}} + \sqrt{\frac{2\log(4\log_2(t+1)/\delta)}{t-s}}\right\}$$

$$= \left\{\exists s \in [1:t], \widehat{\mu}_i(1:s) - \sqrt{\frac{2\log(4\log_2(t+1)/\delta)}{s}} > \widehat{\mu}_i(s+1:t) + \sqrt{\frac{2\log(4\log_2(t+1)/\delta)}{t-s}}\right\}$$

$$\bigcup \left\{\widehat{\mu}_i(1:s) + \sqrt{\frac{2\log(4\log_2(t+1)/\delta)}{s}} < \widehat{\mu}_i(s+1:t) - \sqrt{\frac{2\log(4\log_2(t+1)/\delta)}{t-s}}\right\}$$

We can then redefine the good event $\xi_{i,g}^{chg}(1:t)$ as follows:

$$\xi_{i,g}^{chg}(1:t) := \left\{ \forall s \in [1:t], |\widehat{\mu}_i(1:s) - \mu_i(1:s)| \leq \sqrt{\frac{2\log(4\log_2(t+1)/\delta)}{s}} \right\}$$

$$\bigcup \left\{ \forall s \in [1:t], |\widehat{\mu}_i(s+1:t) - \mu_i(s+1:t)| \leq \sqrt{\frac{2\log(4\log_2(t+1)/\delta)}{s}} \right\}. \tag{28}$$

We can then define the bad event from eq. (28) as follows:

$$\overline{\xi_{i,g}^{chg}}(1:t_{c_g}^i:t) := \left\{ \exists s \in [1:t], |\widehat{\mu}_i(1:s) - \mu_i(1:s)| > \sqrt{\frac{2\log(4\log_2(t+1)/\delta)}{s}} \right\}$$

$$\bigcup \left\{ \exists s \in [1:t], |\widehat{\mu}_i(s+1:t) - \mu_i(s+1:t)| > \sqrt{\frac{2\log(4\log_2(t+1)/\delta)}{s}} \right\}. \tag{29}$$

Next we bound the first event of the inequality in eq. (29) in the same way as Lemma 1 as follows:

$$\mathbb{P}\left( \exists s \geq 1, \widehat{\mu}_i(1:s) - \mu_i(1:s) \geq \sqrt{\frac{2\log(4\log_2(t+1)/\delta)}{s}} \right) \leq \frac{\delta}{4} \tag{30}$$

$$\mathbb{P}\left( \exists s \geq 1, \widehat{\mu}_i(1:s) - \mu_i(1:s) \leq -\sqrt{\frac{2\log(4\log_2(t+1)/\delta)}{s}} \right) \leq \frac{\delta}{4}. \tag{31}$$

Similarly, we can show that the second event of eq. (29) the inequality,

$$\mathbb{P}\left( \exists s \geq 1, \widehat{\mu}_i(s+1:t) - \mu_i(s+1:t) \geq \sqrt{\frac{2\log(4\log_2(t+1)/\delta)}{t-s}} \right) \leq \frac{\delta}{4} \tag{32}$$

$$\mathbb{P}\left( \exists s \geq 1, \widehat{\mu}_i(s+1:t) - \mu_i(s+1:t) \leq -\sqrt{\frac{2\log(4\log_2(t+1)/\delta)}{t-s}} \right) \leq \frac{\delta}{4}. \tag{33}$$

Combining the equations (30), (31), (32) and (33) we get that,

$$\mathbb{P}\left( \overline{\xi_{i,g}^{chg}}(1:t) \right) \leq \delta.$$

It follows then $\mathbb{P}\left( \xi_{i,g}^{chg}(1:t) \right) \geq 1 - \delta$. The claim of the lemma follows. $\square$

### A.5.4 Critical Number of samples for Changepoint Detection

**Lemma 4.** *Let $\widehat{\mu}_i(1:s)$ be the empirical mean of $s$ i.i.d. observations with mean $\mu_i(1:s) =: \mu_{i,g-1}$, and $\widehat{\mu}_i(s+1:t)$ be the empirical mean of $(t-s)$ i.i.d. observations with mean $\mu_i(s+1:t) =: \mu_{i,g}$. Let $\Delta_{i,g}^{chg} := |\mu_{i,g-1} - \mu_{i,g}|$ and the changepoint at $s+1$ round as $t_{c_g}^i$. Define the changepoint event $\xi_{i,g}^{chg}(1:t)$ as in Lemma 3. Then the expected number of times the sub-optimal arm $i$ is sampled before a changepoint $t_{c_g}^i$ is detected is given by*

$$\mathbb{E}[N_i(1:t)] \leq N_{i,g}^{chg}(1:t) + \sum_{s=N_{i,g}^{chg}(1:t)+1}^{t} \mathbb{P}\left( \overline{\xi_{i,g}^{chg}}(s:t) \right)$$

*where,* $N_{i,g}^{chg}(1:t) = \dfrac{8\log(4\log_2(t+1)/\delta)}{\left(\Delta_{i,g}^{chg}\right)^2}.$

*Proof.* Note that the changepoint $t_{c_g}^i$ lie between the round $1$ and $t$. Also note that the mean of the arm $i$ for the rounds $1:s$ is given by $\mu(1:s) =: \mu_{i,g-1}$ and for the rounds $s+1:t$ is given by $\mu(s+1:t) =: \mu_{i,g}$. Then for a sub-optimal arm $i$ it is possible to detect the changepoint at some round $s \in [1:t]$ under the following two events

$$\{\widehat{\mu}_i(1:s) - \beta_i(1:s,\delta) > \widehat{\mu}_i(s+1:t) + \beta_i(s+1:t,\delta)\}, \quad \{\widehat{\mu}_i(1:s) + \beta_i(1:s,\delta) < \widehat{\mu}_i(s+1:t) - \beta_i(s+1:t,\delta)\}. \tag{34}$$

Again note that the events in eq. (34) is not possible if the following events holds true:

$$\left\{\left|\widehat{\mu}_i(1:s) - \widehat{\mu}_i(s+1:t)\right| \leq \sqrt{\frac{2\log(4\log_2(t+1)/\delta)}{s}} + \sqrt{\frac{2\log(4\log_2(t+1)/\delta)}{t-s}}\right\},$$

$$\text{and,} \quad |\mu_{i,g-1} - \mu_{i,g}| < \max\{\beta_i(1:s,\delta), \beta_i(s+1:t,\delta)\}. \tag{35}$$

Now recall from Lemma 1 that at round $t$, the event $\left(\xi_i^{chg}(1:t)\right)$ holds with $1-\delta$ probability. We define the changepoint stopping time $\tau_{c_g}$ as follows:

$$\tau_{c_g} = \min\left\{t : \exists s \in [1,t], |\widehat{\mu}_i(1:s) - \widehat{\mu}_i(s+1:t)| \leq \sqrt{\frac{2\log(4\log_2(t+1)/\delta)}{s}} + \sqrt{\frac{2\log(4\log_2(t+1)/\delta)}{t-s}}\right\}$$

It follows that $\tau_{c_g}$ is $\mathcal{F}_{t-1}$ measurable. Also note that that the second event in (35) is not possible for

$$|\mu_{i,g-1} - \mu_{i,g}| < \max\{\beta_i(1:s,\delta), \beta_i(s+1:t,\delta)\} \implies \Delta_{i,g}^{chg} < 2\sqrt{\frac{2\log(4\log_2(t+1)/\delta)}{N_{i,g}^{chg}(1:t)}}$$

$$\implies N_{i,g}^{chg}(1:t) > \frac{8\log(4\log_2(t+1)/\delta)}{(\Delta_i^{chg})^2}.$$

Let $N_{i,g}^{chg}(1:t)$ be the total number of samples of arm $i$ before the changepoint $t_{c_g}^i$ is detected. Then we can bound the expected number of pulls of $N_{i,g}^{chg}(1:t)$ as follows:

$$\mathbb{E}[N_{i,g}^{chg}(1:t)] = \mathbb{E}\left[\sum_{s=1}^{N_{i,g}^{chg}(1:t)} \mathbf{1}\{\tau_{c_g} \leq N_{i,g}^{chg}(1:t)\}\right] + \mathbb{E}\left[\sum_{s=N_{i,g}^{chg}(1:t)+1}^{t} \mathbf{1}\{\tau_{c_g} > N_{i,g}^{chg}(1:t)\}\right]$$

$$\leq N_{i,g}^{chg}(1:t) + \sum_{s=N_{i,g}^{chg}(1:t)+1}^{t} \mathbb{P}\left(\left\{\tau_{c_g} > N_{i,g}^{chg}(1:t)\right\}\right)$$

$$\leq N_{i,g}^{chg}(1:t) + \sum_{s=N_{i,g}^{chg}(1:t)+1}^{t} \mathbb{P}\left(\left\{|\widehat{\mu}_i(1:s) - \widehat{\mu}_i(s+1:t)| > \sqrt{\frac{2\log(4\log_2(t+1)/\delta)}{s}} + \sqrt{\frac{2\log(4\log_2(t+1)/\delta)}{t-s}}\right\}\right)$$

$$\leq N_{i,g}^{chg}(1:t) + \sum_{s=N_{i,g}^{chg}+1}^{t} \mathbb{P}\left(\overline{\xi_{i,g}^{chg}(s:t)}\right).$$

The claim of the lemma follows. $\qquad\square$

### A.5.5 Total Number of Samples of Baseline Arm

**Lemma 5.** *Let us consider a single time segment $\rho_g$ starting from round $1$ till round $t$. When the optimality event $\xi^{opt}(1:t)$ holds with high probability $1-\delta$ in $\rho_g$ then the total pulls of the baseline arm is bounded by*

$$N_0^{bse}(1:t) \leq \frac{1}{\alpha\mu_{0,g}} \sum_{i\in[K]} \frac{16\log(4\log_2(\tau+1)/\delta)}{\max\{\Delta_{i,g}^{opt}, \Delta_{0,g}^{opt} - \Delta_{i,g}^{opt}\}}.$$

*Proof.* We first define the estimated budget as follows:

$$\widehat{Z}(1:t) := \sum_{s=1}^{t-1} L_{I_s}(1:s) + L_{u_t}(1:t) - (1-\alpha)tU_0(1:t-1)$$

$$= \sum_{i=1}^{K} N_i(1:t-1)L_i(1:t-1) + L_{u_t}(1:t) + N_0(1:t-1)U_0(1:t-1) - (1-\alpha)tU_0(1:t-1) \tag{36}$$

Now note that in eq. (36) if $N_0(1 : t - 1) < (1 - \alpha)t$, then the last term is negative and $U_0(1 : t) \geq \mu_0(1 : t)$. Conversely, if $N_0(1 : t - 1) \geq (1 - \alpha)t$ then the constraint is satisfied as:

$$\sum_{s=1}^{t} \mu_{I_s}(1 : t) \geq N_0(1 : t - 1)\mu_0(1 : t - 1) \geq (1 - \alpha)\mu_0(1 : t)t$$

Let $\tau_0^{bse}(1 : t) := \max\{t' \in [1 : t] : I_{t'} = 0, \widehat{Z}(1 : t' - 1) < 0\}$ be the last round the baseline arm is pulled. Note that at $\tau_0^{bse}(1 : t) - 1$ the safety budget $\widehat{Z}(\tau_{0,s}^{bse}(1 : t) - 1) < 0$. In the following proof we denote $\tau = \tau_0^{bse}(1 : t)$ for notational convenience. It then follows that:

$$\sum_{i=1}^{K} N_i(1 : \tau - 1)L_i(1 : \tau - 1) + L_{u_\tau}(1 : \tau) + N_0(1 : \tau - 1)U_0(1 : \tau - 1) - (1 - \alpha)\tau U_0(1 : \tau) < 0$$

$$\overset{(a)}{\Longrightarrow} \sum_{i=1}^{K} N_i(1 : \tau - 1)L_i(1 : \tau - 1) + N_0(1 : \tau - 1)U_0(1 : \tau - 1) - (1 - \alpha)\tau U_0(1 : \tau) < 0$$

$$\overset{(b)}{\Longrightarrow} \sum_{i=1}^{K} N_i(1 : \tau - 1)L_i(1 : \tau) + N_0(1 : \tau - 1)U_0(1 : \tau)$$

$$- (1 - \alpha)\left(\sum_{i=0}^{K} N_i(1 : \tau - 1) + 1\right)U_0(1 : \tau) < 0$$

$$\Longrightarrow \sum_{i=1}^{K} N_i(1 : \tau - 1)\left[L_i(1 : \tau) - (1 - \alpha)U_0(1 : \tau)\right] - (1 - \alpha)U_0(1 : \tau)$$

$$+ \alpha N_0(1 : \tau - 1)U_0(1 : N_0(1 : \tau) < 0$$

$$\Longrightarrow \underbrace{\alpha N_0(1 : \tau - 1)\left[U_0(1 : \tau)\right]}_{\textbf{Part A}} < \underbrace{\sum_{i=1}^{K} N_i(1 : \tau - 1)\left[(1 - \alpha)U_0(1 : \tau) - L_i(1 : \tau)\right] + (1 - \alpha)U_0(1 : \tau)}_{\textbf{Part B}}$$

where, $(a)$ follows by dropping $L_{u_t}(1 : \tau) > 0$, and $(b)$ follows by introducing

$$\tau = \left(\sum_{i=0}^{K} N_i(1 : \tau - 1) + 1\right).$$

Now note that at the event $\bigcap_{i=1}^{K} \xi_i^{opt}(1 : t)$, $\mu_0(1 : \tau) < U_0(1 : \tau)$. Hence, we can lower bound Part A as follows:

$$\text{Part A} \leq \alpha N_0(1 : \tau - 1)\left[\mu_0(1 : \tau)\right].$$

Similarly we can upper bound Part B as follows:

$$\text{Part B} = \sum_{i=1}^{K} N_i(1 : \tau - 1)\left[(1 - \alpha)U_0(1 : \tau) - L_i(1 : \tau)\right] + (1 - \alpha)U_0(1 : \tau)$$

Now note that at $\bigcap_{i=1}^{K} \xi_i^{opt}(1 : \tau)$ we have

$$U_0(1 : \tau) \leq \mu_0(1 : \tau) + \sqrt{\frac{2\log(4\log_2(\tau + 1)/\delta)}{N_i(1 : \tau)}} \overset{(a)}{\leq} \mu_0(1 : \tau) + \frac{\Delta_0^{opt}(1 : \tau)}{2}$$

where, $(a)$ follows for $N_i(1 : \tau) \geq \dfrac{8\log(4\log_2(\tau + 1)/\delta)}{(\Delta_i^{opt}(1 : \tau))^2}$ with probability greater than $1 - 2\delta$. Then for the Part B we can

show that

$$\text{Part B} \leq \sum_{i \in [K]} N_i(1:\tau-1)\left[(1-\alpha)\left(\mu_0(1:\tau) + \frac{\Delta_0^{opt}(1:\tau)}{2}\right) - L_i(1:\tau)\right] + (1-\alpha)\left(\mu_0(1:\tau) + \frac{\Delta_0^{opt}(1:\tau)}{2}\right)$$

$$\leq \sum_{i \in [K]} N_i(1:\tau-1)\left[(1-\alpha)\left(\mu_0(1:\tau) + \frac{\Delta_0^{opt}(1:\tau)}{2}\right) - \mu_i(1:\tau)\right.$$

$$\left. + \sqrt{\frac{2\log(4\log_2(\tau+1)/\delta)}{N_i(1:\tau-1)}}\right] + (1-\alpha)\left(\mu_0(1:\tau) + \frac{\Delta_0^{opt}(1:\tau)}{2}\right)$$

$$\overset{(a)}{=} \sum_{i \in [K]} N_i(1:\tau-1)c_i + \sqrt{N_i(1:\tau-1)\left(2\log(4\log_2(\tau+1)/\delta)\right)} + (1-\alpha)\left(\mu_0(1:\tau) + \frac{\Delta_0^{opt}(1:\tau)}{2}\right)$$

$$\overset{(b)}{\leq} \sum_{i \in [K]} \frac{8c_i\log(4\log_2(\tau+1)/\delta)}{(\Delta_i^{opt}(1:\tau))^2} + \frac{8\log(4\log_2(\tau+1)/\delta)}{(\Delta_i^{opt}(1:\tau))} + (1-\alpha)\left(\mu_0(1:\tau) + \frac{\Delta_0^{opt}(1:\tau)}{2}\right)$$

where, in $(a)$ we define $c_i = (1-\alpha)\left(\mu_0(1:\tau) + \frac{\Delta_0(1:\tau)}{2}\right) - \mu_i(1:\tau)$, and $(b)$ follows by setting $N_i(1:\tau-1) \geq \frac{8\log(4\log_2(\tau+1)/\delta)}{(\Delta_i^{opt}(1:\tau))^2}$. Now for $c_i \geq 0$ we can show that

$$(1-\alpha)(\mu_0(1:\tau) + \frac{\Delta_0^{opt}(1:\tau)}{2}) - \mu_i(1:\tau) \geq 0$$

$$\implies \mu_{i^*}(1:\tau) - (1-\alpha)\left(\mu_0(1:\tau) + \frac{\Delta_0^{opt}(1:\tau)}{2}\right) \leq \mu_{i^*}(1:\tau) - \mu_i(1:\tau)$$

$$\implies \Delta_i(1:\tau) \geq (1+\alpha)\frac{\Delta_0^{opt}(1:\tau)}{2} + \alpha(\mu_0(1:\tau))$$

Again for $c_i < 0$ we can show that

$$(1-\alpha)(\mu_0(1:\tau) + \frac{\Delta_0^{opt}(1:\tau)}{2}) - \mu_i(1:\tau) < 0$$

$$\implies \mu_{i^*}(1:\tau) - (1-\alpha)\left(\mu_0 + \frac{\Delta_0^{opt}(1:\tau)}{2}\right) > \mu_{i^*}(1:\tau) - \mu_i(1:\tau)$$

$$\implies \Delta_i(1:\tau) < (1+\alpha)\frac{\Delta_0^{opt}(1:\tau)}{2} + \alpha(\mu_0(1:\tau))$$

Combining everything we can show that

$$\alpha N_0(1:\tau-1)\left[\mu_0(1:\tau)\right] \leq \sum_{i \in [K]} \frac{8c_i\log(4\log_2(\tau+1)/\delta)}{(\Delta_i^{opt}(1:\tau))^2} + \frac{8\log(4\log_2(\tau+1)/\delta)}{(\Delta_i^{opt}(1:\tau))} + (1-\alpha)\left(\mu_0(1:\tau) + \frac{\Delta_0^{opt}(1:\tau)}{2}\right)$$

$$N_0(1:\tau) \overset{(a)}{\leq} \frac{1}{\alpha\mu_0(1:\tau)}\sum_{i \in [K]} \frac{16\log(4\log_2(\tau+1)/\delta)}{\max\{\Delta_i^{opt}(1:\tau), 0.5(1-\alpha)\Delta_0^{opt}(1:\tau) + \alpha\mu_0(1:\tau) - \Delta_i^{opt}(1:\tau)\}}$$

$$N_0(1:\tau) \leq \frac{1}{\alpha\mu_0(1:\tau)}\sum_{i \in [K]} \frac{16\log(4\log_2(\tau+1)/\delta)}{\max\{\Delta_i^{opt}(1:\tau), \Delta_0^{opt}(1:\tau) - \Delta_i^{opt}(1:\tau)\}}$$

where, $(a)$ follows as $\alpha(\mu_0(1:t) - B) \geq \frac{\alpha(\mu_0(1:t) - B)}{2}$. Now considering the $g$-th changepoint we can show that

$$N_0^{bse}(1:\tau) = N_{0,g}^{opt} = \frac{1}{\alpha\mu_{0,g}}\sum_{i \in [K]} \frac{16\log(4\log_2(\tau+1)/\delta)}{\max\{\Delta_{i,g}^{opt}, \Delta_{0,g}^{opt} - \Delta_{i,g}^{opt}\}}.$$

The claim of the lemma follows. $\square$

## A.6  LOWER BOUND IN SAFE GLOBAL CHANGEPOINT SETTING

**Theorem 3.** *(Restatement) Let $\mathcal{E}, \overline{\mathcal{E}}$ be two bandit environment and there exits a global changepoint at $t_{c_1} = T/2$. Let $\alpha > 0$ be the safety parameter and $\mu_{0,\min}$ be the mean of the minimum safety mean over the changepoint segments. Then the lower bound is given by*

$$\mathbb{E}_{\mathcal{E},\overline{\mathcal{E}}}[R_T] \geq \left\{ \frac{K}{(16e+8)\alpha\mu_{0,\min}} + \frac{\log T}{\alpha\mu_{0,\min}}, \frac{\sqrt{KT}}{\sqrt{32e+16}} + \frac{\log T}{\alpha\mu_{0,\min}} \right\}.$$

**Step 1 (Safe Risk):** Consider a setting with a single changepoint $g$. For the segment $\rho_0$ from $t = 1$ to $t_{c_1} - 1$ define the environment $\mathcal{E} \in [0,1]^K$ such that $\mu_i(1 : t_{c_1} - 1) = \mu_0(1 : t_{c_1} - 1) - \Delta$ for all $i \in [K]$. We assume that $\mu_0(1 : t_{c_1} - 1)$ and $\Delta$ are such that $\mu_i(1 : t_{c_1} - 1) \geq 0$. Define environment $\mathcal{E}^{(i)}$ for each $i = 1, \ldots, K$ by

$$\mathcal{E}_j^{(i)} = \begin{cases} \mu_0(1 : t_{c_1} - 1) + \Delta, & \text{for } j = i \\ \mu_0(1 : t_{c_1} - 1) - \Delta, & \text{otherwise.} \end{cases}$$

Similarly for the segment $\rho_1$ from $t_{c_1}$ to $T$ define the environment $\overline{\mathcal{E}} \in [0,1]^K$ such that $\mu_i(t_{c_1} : T) = \mu_0(t_{c_1} : T) - \Delta$ for all $i \in [K]$. Again, we assume that $\mu_0(t_{c_1} : T)$ and $\Delta$ are such that $\mu_i(t_{c_1} : T) \geq 0$. Define environment $\overline{\mathcal{E}}^{(i)}$ for each $i = 1, \ldots, K$ by

$$\overline{\mathcal{E}}_j^{(i)} = \begin{cases} \mu_0(t_{c_1} : T) + \Delta', & \text{for } j = i \\ \mu_0(t_{c_1} : T) - \Delta', & \text{otherwise.} \end{cases}$$

where, $\Delta' > 0$. Note that $\mathcal{E}$ and $\mathcal{E}^{(i)}$ differ only in the $i$-th component: $\mu_i(1 : t_{c_1} - 1) = \mu_0(1 : t_{c_1} - 1) - \Delta$ whereas $\mu_i^{(i)}(1 : t_{c_1} - 1) = \mu_0(1 : t_{c_1} - 1) + \Delta$. Then the KL divergence between the reward distributions of the $i$ th arms is given by

$$\mathrm{KL}\left(\mathcal{E}_i, \mathcal{E}_i^{(i)}\right) = (2\Delta)^2/2 = 2\Delta^2$$

Using Theorem 9 of [Wu et al., 2016] (see Proposition 3) we can show that in $\mathcal{E}$ the lower bound to the safe regret is given by

$$\mathbb{E}_{\mathcal{E}}[R_{t_{c_1}}] \geq \begin{cases} \frac{\sqrt{Kt_{c_1}}}{\sqrt{16e+8}}, & \text{when } \alpha \geq \frac{\sqrt{K}}{\mu_0(1:t_{c_1}-1)\sqrt{(16e+8)t_{c_1}}} \\ \frac{K}{(16e+8)\alpha\mu_0(1:t_{c_1}-1)}, & \text{otherwise} \end{cases} \tag{37}$$

and for $\overline{\mathcal{E}}$ the lower bound is given by

$$\mathbb{E}_{\overline{\mathcal{E}}}[R_{T-t_{c_1}}] \geq \begin{cases} \frac{\sqrt{K(T-t_{c_1})}}{\sqrt{16e+8}}, & \text{when } \alpha \geq \frac{\sqrt{K}}{\mu_0(t_{c_1}:T)\sqrt{(16e+8)(T-t_{c_1})}} \\ \frac{K}{(16e+8)\alpha\mu_0(t_{c_1}:T)}, & \text{otherwise} \end{cases} \tag{38}$$

**Step 2 (Changepoint risk):** Now we introduce the changepoint detection framework. In this framework define a false alarm rate constraint specified by a tuple $(m, \alpha)$, where $m \in \mathbb{Z}^+$ denotes an a priori fixed time such that the probability of an admissible change detector stopping before time $m$ is at most $\delta \in (0,1)$, namely, $\mathbb{P}[\tau < m] \leq \delta$. This framework is similar to what defined in [Gopalan et al., 2021]. Then we know from Theorem 2 of [Gopalan et al., 2021] (see Proposition 5) that for any bandit changepoint algorithm satisfying $\mathbb{P}[\tau < m] \leq \delta$, we have

$$\mathbb{E}_{\mathcal{E},\overline{\mathcal{E}}}[\tau] \geq \min\left\{ \frac{\frac{1}{20}\log\frac{1}{\delta}}{\max_{i\in[K]}\mathrm{KL}\left(\mathcal{E}_i, \overline{\mathcal{E}}_i\right)}, \frac{m}{2} \right\} = \min\left\{ \frac{\frac{1}{10}\log\frac{1}{\delta}}{\underline{\Delta}^2}, \frac{m}{2} \right\}. \tag{39}$$

where $\underline{\Delta} > 0$ is the minimum changepoint gap between any arms in $\mathcal{E}, \overline{\mathcal{E}}$.

**Step 3:** Combining the two steps above and using (37), (38), and (39) we can show that the total regret lower bound is

$$
\mathbb{E}_{\mathcal{E},\overline{\mathcal{E}}}[R_T] \geq
\begin{cases}
\mathbb{E}_{\overline{\mathcal{E}}}[R_{t_{c_1}}] + \mathbb{E}_{\overline{\mathcal{E}}}[R_{T-t_{c_1}}] + \underline{\Delta}\mathbb{E}_{\mathcal{E},\overline{\mathcal{E}}}[\tau], & \text{when } \alpha \geq \frac{\sqrt{K}}{\mu_0(1:t_{c_1}-1)\sqrt{(16e+8)t_{c_1}}} \\[2mm]
\mathbb{E}_{\overline{\mathcal{E}}}[R_{t_{c_1}}] + \mathbb{E}_{\overline{\mathcal{E}}}[R_{T-t_{c_1}}] + \underline{\Delta}\mathbb{E}_{\mathcal{E},\overline{\mathcal{E}}}[\tau], & \text{otherwise}
\end{cases}
$$

$$
\implies \mathbb{E}_{\mathcal{E},\overline{\mathcal{E}}}[R_T] \geq
\begin{cases}
\frac{\sqrt{K(T-t_{c_1})}}{\sqrt{16e+8}} + \frac{\sqrt{K(T-t_{c_1})}}{\sqrt{16e+8}} + \min\left\{ \frac{1}{10}\log\frac{1}{\delta}, \frac{m\underline{\Delta}}{2} \right\}, & \text{when } \alpha \geq \frac{\sqrt{K}}{\mu_0(1:t_{c_1}-1)\sqrt{(16e+8)t_{c_1}}} \\[2mm]
\frac{K}{(16e+8)\alpha\mu_0(1:t_{c_1}-1)} + \frac{K}{(16e+8)\alpha\mu_0(t_{c_1}:T)} + \min\left\{ \frac{1}{10}\log\frac{1}{\delta}, \frac{m\underline{\Delta}}{2} \right\}, & \text{otherwise}
\end{cases}
$$

$$
\implies \mathbb{E}_{\mathcal{E},\overline{\mathcal{E}}}[R_T] \geq
\begin{cases}
\frac{\sqrt{K(T-t_{c_1})}}{\sqrt{16e+8}} + \frac{\sqrt{K(T-t_{c_1})}}{\sqrt{16e+8}} + \frac{\log T}{\alpha\mu_{0,\min}}, & \text{when } \alpha \geq \frac{\sqrt{K}}{\mu_0(1:t_{c_1}-1)\sqrt{(16e+8)t_{c_1}}} \\[2mm]
\frac{K}{(16e+8)\alpha\mu_0(1:t_{c_1}-1)} + \frac{K}{(16e+8)\alpha\mu_0(t_{c_1}:T)} + \frac{\log T}{\alpha\mu_{0,\min}}, & \text{otherwise}
\end{cases}
$$

where, $(a)$ follow by setting $\underline{\Delta} = \alpha\mu_{0,\min}$, $m = T$ as the last time the changepoint needs to be detected, and $\delta = \frac{1}{T}$. Combining everything and setting $t_{c_1} = T/2$ we can show that

$$
E_{\mathcal{E},\overline{\mathcal{E}}}[R_T] \geq \left\{ \frac{K}{(16e+8)\alpha\mu_{0,\min}} + \frac{\log T}{\alpha\mu_{0,\min}}, \frac{\sqrt{KT}}{\sqrt{32e+16}} + \frac{\log T}{\alpha\mu_{0,\min}} \right\}.
$$

## A.7 ADDITIONAL EXPERIMENT DETAILS

**Setting for choosing different $\alpha$:** In this experiment we illustrate the tension between the risk parameter $\alpha$ and the cumulative regret for various values of $\alpha$. The LCS environment consist of 4 arms (including baseline) and the evolution of means are shown in Figure 4 (Left). We plot the regret versus various values of $\alpha \in [0, 1]$ in Figure 4 (Right). We see that SLR has high regret in high risk ($\alpha \to 0$) setting. However its performance lies between D-UCB and GLR-UCB which are safety oblivious algorithms. SLR outperforms both UMOSS and safety aware CUCB algorithm. We see that CUCB only performs well when risk is low ($\alpha \to 1$). This is the changepoints occur too fast and when ($\alpha \to 0$) the CUCB is forced to choose the baseline arm always.

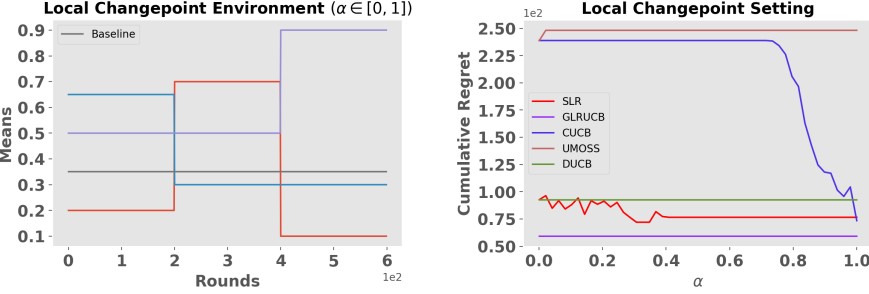

Figure 4: (Left) Local changepoint environment with $T = 1500$, $K^+ = 4$ and changepoints at $t = 500, 1000$ and $1500$. Again note that some arms do not change at these changepoints. (Right) Shows the regret vs. $\alpha \in [0, 1]$ plot.

**Remark 1.** *In all the environments we tested against adversarial CUCB as recommended in Wu et al. [2016] for adversarial setting. The adversarial CUCB again always samples the baseline arm as it is not suited for the safety constraint* (1) *and achieves linear regret similar to CUCB. So we omit it's plot from the figures.*

**Time Complexity** Note that Ad-Switch suffers an expensive time complexity of $\Theta(KT^4)$ for performing the changepoint detection test whereas GLR-UCB , UCB-CPD , M-UCB , SGR , SLR suffers a time complexity of $\Theta(KT^2)$.

**Algorithmic Implementations:** We implement the following algorithms:

1) **GLR-UCB [Besson and Kaufmann, 2019]:** This algorithm is an actively adaptive changepoint detection algorithm. It uses the GLRT statistic to detect changepoints and restart. It is implemented with the Gaussian KL function with known variance to calculate the GLRT statistic. It works for both GCS and LCS environment and requires the horizon $T$ as input.

2) **UCB-CPD [Mukherjee and Maillard, 2019]:** This algorithm is also an actively adaptive changepoint detection algorithm. It uses confidence based scan statistic and we use the $\beta_i(t, \delta)$ in (3) to implement its changepoint detector as opposed to the time-uniform bound used by Mukherjee and Maillard [2019]. The performance difference is negligible for such a short horizon. It requires the horizon $T$ as input.

3) **UCB-CPDE :** This algorithm is also an actively adaptive changepoint detection algorithm. UCB-CPDE is similar to UCB-CPD but suited for LCS as it conducts forced exploration of arms. We again use the same $\beta_i(t, \delta)$ in (3) to implement its changepoint detector. It also requires the horizon $T$ as input.

4) **D-UCB [Garivier and Cappé, 2011]:** This is a passive algorithm that does not detect the changepoints and restart. We set the exploration parameter $\gamma = 1 - \frac{1}{4}\sqrt{\frac{1}{T}}$ as recommended in Garivier and Cappé [2011].

5) **CUCB [Wu et al., 2016]:** This is the safety aware algorithm for the stochastic bandit setting. It uses the new safety constraint in (1) instead of the constraint in Wu et al. [2016] (see Proposition 2). So it calculates the new empirical safety budget $\widehat{Z}(1:t)$ (7) and when the safety budget is positive it uses the UCB-index [Auer et al., 2002a] to sample an arm, otherwise it samples the baseline.

6) **Adversarial CUCB [Wu et al., 2016]:** This is a safety aware algorithm for the adversarial bandit setting. It uses the new safety constraint in (1) instead of the constraint in Wu et al. [2016]. So it calculates the new empirical safety budget $\widehat{Z}(1:t)$ (7) and when the safety budget is positive it uses the EXP3 [Auer et al., 2002b] to sample an arm, otherwise it samples the baseline.

7) **UMOSS [Lattimore, 2015]:** The Unbalanced Moss (UMOSS ) is a conservative exploration bandit. The UMOSS is tuned with the parameter

$$\tilde{B}_0 = \frac{TK}{\sqrt{TK} + \frac{K}{\alpha\mu_0}}, \quad \tilde{B}_i = \tilde{B}_K = \sqrt{TK} + \frac{K}{\alpha\mu_0}.$$

The quantity $\tilde{B}_i$ determines the regret of UMOSS with respect to arm $i$ up to constant factors, and must be chosen to lie inside the Pareto frontier given by [Lattimore, 2015]. UMOSS has no guarantees for the safety constraint (1). It was found to perform comparably to the highly constrained CUCB in Wu et al. [2016]. UMOSS requires $\tilde{B}_0, \ldots, \tilde{B}_K$ as inputs and so it requires the knowledge of the mean of the baseline.

## A.8    TABLE OF NOTATIONS

| Notation | Definition |
|:---:|:---:|
| $K$ | Total number of arms |
| $T$ | Horizon |
| $\delta$ | Probability of error |
| $\beta_i(1:t,\delta)$ | $\sqrt{2\log(4\log_2(t+1)/\delta)/N_i(1:t)}$ |
| $B(T,\delta)$ | $16\log(4\log_2(T/\delta))$ |
| $\alpha$ | Risk Parameter |
| $G_T$ | Total Changepoints till horizon $T$ |
| $I_s$ | arm sampled at round $s$ |
| $1:t$ | Rounds from $1$ to $t$ |
| $\rho_g$ | $t_{c_g}:t_{c_{g+1}}-1$ |
| $\Delta_{i,g}^{opt}$ | $\mu_{i^*,g}-\mu_{i,g}$ for segment $\rho_g$ |
| $\Delta_{i,g}^{chg}$ | $\lvert\mu_{i,g}-\mu_{i,g+1}\rvert$ for segment $\rho_g$ |
| $H_{i,g-1}^{(1)}$ | $\max\left\{\dfrac{1}{\Delta_{i,g-1}^{opt}},\dfrac{\Delta_{i,g-1}^{opt}}{\left(\Delta_{i,g-1}^{chg}\right)^2}\right\}$ |
| $H_{i,g}^{(2)}$ | $\max_{j\in[K]^+}\dfrac{\Delta_{i,g}^{opt}}{\left(\Delta_{j,g}^{chg}\right)^2}$ |
| $\overline{H_{i,g}^{(2)}}$ | $\dfrac{\Delta_{i,g}^{opt}}{\left(\Delta_{i,g}^{chg}\right)^2}$ |
| $H_{i,g}^{(3)}$ | $\dfrac{\Delta_{\max,g}^{opt}}{\max\{\Delta_{i,g}^{opt},\Delta_{0,g}^{opt}-\Delta_{i,g}^{opt}\}}$ |
| $N_{i,g}^{opt}$ | Critical number of samples for Optimality Detection at $\rho_g$ |
| $N_{i,g}^{chg}$ | Critical number of samples for Changepoint Detection at $\rho_g$ |
| $N_{0,g}^{bse}$ | Total number of samples for baseline at $\rho_g$ |

Table 1: Table of Notations