# OpenReview forum: "Safety Aware Changepoint Detection for Piecewise i.i.d. Bandits"
_auai.org/UAI/2022/Conference — UAI 2022 Poster_

### Official Review · Reviewer_ZN7d · 2022-03-21

**Q2(1) Originality/Novelty:** 3
**Q2(2) Significance/Impact:** 3
**Q2(3) Correctness/Technical Quality:** 3
**Q2(6) Clarity Of Writing:** 3
**Q6 Overall Score:** 6
**Q8 Confidence In Your Score:** 3

**Q1 Summary And Contributions:**

This work studies the safe version of the piecewise stochastic MAB problem. They propose two algorithms that can detect global or local changepoints and satisfy the safety constraint simultaneously. The authors give the corresponding regret upper bounds for these two methods, and show that their results almostly match the lower bound, up to log factors of T. Finally, the authors carry on experiments to show their methods' effectiveness and safety against several existing works.

**Q2 Assessment Of The Paper:**

More detailed information regarding each of these aspects is given below:

**Q2(4) Quality Of Experiments (Optional):**

2: Fair: The experimental evaluation is weak: important baselines are missing, or the results do not adequately support the main claims.

**Q2(5) Reproducibility:**

3: Good: Key resources (e.g., proofs, code, data) are available and key details (e.g., proofs, experimental setup) are sufficiently well-described for competent researchers to confidently reproduce the main results.

**Q3 Main Strengths:**

1. The problem is new. Although it is a combination of two existing settings, the combination is not orthogonal since there exists a tradeoff between catching up with the changing environment and satisfying conservative safety constraints, which is an interesting problem to investigate on.
2. The solution is complete. Although the solution is not surprising since most techniques are from existing works, it can deal with the current problem well.
3. Sound theoretical guarantees.
4. Empirical evaluations.

**Q4 Main Weakness:**

See Q5.

**Q5 Detailed Comments To The Authors:**

1. Question on the safety budget: Are there any theoretical guarantees on the empirical safety budget, like the safety constraints are satisfied with high probability? I guess that there exist such kinds of guarantees since intuitively the empirical budget consists of UCBs and LCBs, which is a pretty conservative judgment, however, in my point of view, the authors should make a formal statement of this.
2. Question on global/local CD:
    1) since the global setting is a special case of the local one, why the former should be considered individually?
    2) in my point of view, the main difference between the two settings is that the local method includes a forced-exploration operation, which is crucial since there exists an exploration-exploitation tradeoff. Can the authors further explain how to balance this tradeoff (forced exploration or pull the UCB arm) in the local CD setting?
3. The experiment part:
    1) the most crucial issue is that there are no evaluations on how much the safety constraint is violated. This problem actually has 2 optimization objectives, regret minimization and safety. All the experiment results only show the performance on regret minimization, and the performance on safety is only given via some statement, which is not convincing for me.
    2) the time efficiency: the CPD algorithm seems not to be time-efficient since in the worst-case the time complexity will be $O(T^2)$. Do the authors have some comments on this issue?
    3) Please use vector figures (e.g., pdf) to present the results.
4. There are undesired spaces after the methods' name, e.g., SLR, SGR, etc.


**Q7 Justification For Your Score:**

See Q3.

**Q9 Complying With Reviewing Instructions:**

1: Yes.

---

### Official Review · Reviewer_stwp · 2022-04-12

**Q2(1) Originality/Novelty:** 3
**Q2(2) Significance/Impact:** 2
**Q2(3) Correctness/Technical Quality:** 2
**Q2(6) Clarity Of Writing:** 3
**Q6 Overall Score:** 7
**Q8 Confidence In Your Score:** 4

**Q1 Summary And Contributions:**

The authors consider stochastic bandits with a finite number of changes under a constraint that the cumulative reward should be above a constant factor of the default action reward.  A distinction is made between global changes (affecting all the arms) and local changes (affecting some arms). The authors propose adaptive algorithms which detect the changes and provide corresponding regret bounds. Experimental results on synesthetic data and real world datasets validate the theoretical results.

**Q2 Assessment Of The Paper:**

More detailed information regarding each of these aspects is given below:

**Q2(4) Quality Of Experiments (Optional):**

3: Good: The experimental evaluation is adequate, and the results convincingly support the main claims.

**Q2(5) Reproducibility:**

3: Good: Key resources (e.g., proofs, code, data) are available and key details (e.g., proofs, experimental setup) are sufficiently well-described for competent researchers to confidently reproduce the main results.

**Q3 Main Strengths:**

1. In my opinion, the considered problem is analytically challenging and practically relevant.

2. The authors have treated this problem extensively and the given experimental results appear to be comprehensive.

**Q4 Main Weakness:**

1. Some of the mathematical analysis are imprecisely described.

2. The description of how a border case is handled in the analysis is unclear.

See answer to Q5 for more details.

**Q5 Detailed Comments To The Authors:**

1. The decomposition given in 12 is described to be divided into $\rho_g$ where each $\rho_g$ is further divided into rounds before $t_{c_g}$ and rounds before detection of $t_{c_g}$. Doesn't this contradict with the definition of $\rho_g$ given in the beginning of section 3 which indicates that $\rho_g$ starts at $t_{c_g}$ (so there are no rounds in $\rho_g$ before $t_{c_g}$)? I believe the decomposition given in 12 is not into $\rho_g$'s but rather into $\tau_{c_{g-1}} : \tau_{c_g}-1$ i.e. the time interval between the detection of change $g-1$ and the detection of change $g$. This interval is further divided into before and after the round at which the g'th change occurs (i.e. $t_{c_g}$).

2. Continuing on the above point, if the decomposition is indeed in terms of the intervals between the detection of change points, then the summation should be on the number of detected change points which might not be equal to the number of actual change points $G_T$. Currently the summation is on the $G_T$.

3. I also do not see where the case of false detection of a change point is handled.

4. Paragraph 2 in the introduction mentions that "(t)he baseline arm represents expert’s belief over the current user preferences and may change over time". Unless I am misreading the above statement, it allows the identity of the baseline arm to change over time which I believe is a practically relevant scenario. Yet the identity of the baseline arm is assumed to remain the same over time throughout the paper.

5. The definition of the empirical safety budget given in (7) contains an upper bound of the UCB arm $u_t$ while in other papers dealing with similar constraint (e.g. Wu et al. (2016)) the corresponding quantity contains the lower bound of the UCB arm. Could you explain the reasoning behind this choice?

6. Minor comments about presentation: Please indent algorithm 2 to be more readable. The current presentation makes it hard to read,

**Q7 Justification For Your Score:**

I believe the paper studies an important problem setting and the presented algorithms are a natural solution. I will consider increasing the score if my concerns stated in the answer to Q6 are justifiably answered.

*****************Post-rebuttal*****************
The rebuttal has clarified the main issues raised in my review so I raise my score.

**Q9 Complying With Reviewing Instructions:**

1: Yes.

---

### Official Review · Reviewer_74Nw · 2022-04-16

**Q2(1) Originality/Novelty:** 2
**Q2(2) Significance/Impact:** 2
**Q2(3) Correctness/Technical Quality:** 3
**Q2(6) Clarity Of Writing:** 3
**Q6 Overall Score:** 5
**Q8 Confidence In Your Score:** 3

**Q1 Summary And Contributions:**

This paper studies the safety-aware piecewise i.i.d. bandits under safety constraints.  The main techniques to deal with changing feedback is to detect the change and restart. This paper investigates a cumulative reward safety condition and two actively adaptive algorithms which satisfy the safety constraints, dealing with both local change and global change. The paper provides regret bounds on the algorithms.

**Q2 Assessment Of The Paper:**

More detailed information regarding each of these aspects is given below:

**Q2(4) Quality Of Experiments (Optional):**

2: Fair: The experimental evaluation is weak: important baselines are missing, or the results do not adequately support the main claims.

**Q2(5) Reproducibility:**

3: Good: Key resources (e.g., proofs, code, data) are available and key details (e.g., proofs, experimental setup) are sufficiently well-described for competent researchers to confidently reproduce the main results.

**Q3 Main Strengths:**

The setting is a novel one: studying piecewise iid bandits under cumulative reward safety conditions.

The paper seems to be technically sound.


**Q4 Main Weakness:**

Even though the setting is a novel one, it also feels like a combination of two previous well-studied settings: safe bandits and bandit with changing rewards. A large portion of the method is also based on or similar with the previous method. For example, change point detection is a separate subroutine and does not really affect the safe exploration process. It makes me wonder whether this is a carefully crafted setting that simply requires the addition of methods from two fields.

The experiments are not analyzed carefully. For example, the proposed method is not competitive in some settings. But there is no explanation. There could also be more synthetic cases and more real world cases to compare with the baselines.


**Q5 Detailed Comments To The Authors:**

I am a bit confused by the analysis of figure 2: “The adaptive algorithms like UCB-CPD , GLR-UCB perform well as they detect the change points and restart but they do not satisfy the safety constraints. Note that SGR performs similar to GLR-UCB , UCB-CPD as it also detects the change- points and restarts as well as satisfy the safety constraint. ” These two sentences are contradictory to each other, and actually the SGR is not competitive than GLR-UCB , UCB-CPD. Therefore, the claim in the abstract about “our safety-aware algorithms match the performance of the state-of-the-art actively adaptive algorithms that do not satisfy the safety constraint” may not be true or at least need more evidence to support it. Intuitively, there could be a tradeoff between safety and performance. So how the method bypasses this tradeoff is not clear to me.



**Q7 Justification For Your Score:**

This paper studies a novel setting but the contribution can be incremental depending on who you see it. There are some issues with the experiments and also the analysis.



**Q9 Complying With Reviewing Instructions:**

1: Yes.

---

### Official Review · Reviewer_MW4y · 2022-04-17

**Q2(1) Originality/Novelty:** 3
**Q2(2) Significance/Impact:** 3
**Q2(3) Correctness/Technical Quality:** 3
**Q2(6) Clarity Of Writing:** 3
**Q6 Overall Score:** 7
**Q8 Confidence In Your Score:** 3

**Q1 Summary And Contributions:**

The paper studies algorithms for stochastic bandits when (i) arm-means can change abruptly at a few change points, (ii) a safety constraint needs to be satisfied with probability 1. The paper formalizes the problem by making concrete assumptions on the inter change time and the jump between changes. For the stated assumptions, problem dependent complexity parameters are identified to prove a regret upper and lower bound. Simulations show that algorithms are not sensitive to the assumptions.



**Q2 Assessment Of The Paper:**

More detailed information regarding each of these aspects is given below:

**Q2(4) Quality Of Experiments (Optional):**

3: Good: The experimental evaluation is adequate, and the results convincingly support the main claims.

**Q2(5) Reproducibility:**

3: Good: Key resources (e.g., proofs, code, data) are available and key details (e.g., proofs, experimental setup) are sufficiently well-described for competent researchers to confidently reproduce the main results.

**Q3 Main Strengths:**

The key strengths of the paper are

(1) Concrete formulation of a practically important problem.

(2) Identifying problem complexity parameters to provide regret upper and lower bounds.

(3) Simple to implement algorithms that perform well in simulations when the theoretical assumptions are violated.

**Q4 Main Weakness:**

The writing can be improved. See my detailed comments.

**Q5 Detailed Comments To The Authors:**

 I have a few comments regarding the presentation below, which I believe are minor and can be fixed in the write up.

Comments -

(1) The word "restart" is confusing and would be good to formally define the restart operation (for example using the notation described in Line 5 of Algorithm 2). For example, the last two lines of Section 2 defines the notation of $\widehat{\mu}\_i(1:t)  = \frac{\sum\_{s=1}^{t}X\_{I_s}\mathbf{1}(I_s=i)}{\sum\_{s=1}^{t}\mathbf{1}(I_s=i)}$, I am not sure how the equality (a) in Equation (8) follows. Perhaps, by "restart" you mean setting all rewards and arm-pulled information to 0 ?

(2) In Equation (10), there is $i$ missing in the RHS. Concretely, is the test in the RHS hold for all $i\in[K]^{+}$ or $\exists i \in [K]^{+}$. I presume it is the latter. See my point (4) below.

(3) The operation in Line 5 of Algorithm 2 is confusing. If at time $t^{'}$, a change is detected in arm $i \in [K]^{+}$, then Line $5$ is to ensure that at time $t$, the cumulative reward and the number of times every arm is played is set to 0, so that at time $t+1$, is equivalent to the first step of a fresh bandit algorithm ?

(4) From the description in the first two lines in Section 3.3, it seems that the arm-means for all arms in a segment $\rho_g$ are fixed. This is indeed consistent if the test in the RHS of Equation (10) is $\exists i \in [K]^{+}$. However, this is inconsistent with the definition of global change-point given just below Equation (2). It seems to me that the notation below Equation (2) is not necessary ?

**Q7 Justification For Your Score:**

The paper is technically solid and the issues I have are with regards to writing and presentation which I believe can be fixed in the write-up.

**Q9 Complying With Reviewing Instructions:**

1: Yes.

---

### Decision · Program_Chairs · 2022-05-15

**Decision:**

Accept (Poster)

**Comment:**

Meta Review: This paper studies safe piecewise-stationary bandits, where the stationary environment can change and the cumulative reward of the agent needs to stay close to that of the baseline action. The proposed solution is A + B, where

A = detect changes using a change-point detector and reset all statistics upon a change

B = maintain an estimate of the available exploration budget and explore whenever possible

and follows from prior works in the respective areas. All reviewers agreed that this paper is well executed, and my opinion is that being A + B is not necessarily bad. Some writing in the paper was imprecise and this was clarified by the rebuttal. No reviewer had major concerns and I support acceptance.